# DIVERGENCE MINIMIZATION PREFERENCE OPTIMIZATION FOR DIFFUSION MODEL ALIGNMENT

## ABSTRACT

Diffusion models have achieved remarkable success in generating realistic and versatile images from text prompts. Inspired by the recent advancements of language models, there is an increasing interest in further improving the models by aligning with human preferences. However, we investigate alignment from a divergence minimization perspective and reveal that existing preference optimization methods are typically trapped in suboptimal mean-seeking optimization. In this paper, we introduce Divergence Minimization Preference Optimization (DMPO), a novel and principled method for aligning diffusion models by minimizing reverse KL divergence, which asymptotically enjoys the same optimization direction as original RL. We provide rigorous analysis to justify the effectiveness of DMPO, and conduct comprehensive experiments to validate its empirical strength across both human evaluations and automatic metrics. Our extensive results show that diffusion models fine-tuned with DMPO can consistently outperform or match existing techniques, specifically consistently outperforming all baseline models across different base models and test sets, achieving the best PickScore in every case, demonstrating the method's superiority in aligning generative behavior with desired outputs. Overall, DMPO unlocks a robust and elegant pathway for preference alignment, bridging principled theory with practical performance in diffusion models.

## 1 INTRODUCTION

Diffusion models (Ho et al., 2020; Sohl-Dickstein et al., 2015; Song and Ermon, 2019; Song et al., 2021b) have emerged as a leading approach for text-to-image (T2I) generation (Ramesh et al., 2022; Pernias et al., 2024; Ramesh et al., 2021), offering a scalable and robust framework for synthesizing images from natural language descriptions. These models typically adopt a single-stage training approach, learning the data distribution from large-scale web-crawled text-image pairs (Rombach et al., 2022; Podell et al., 2023; Peebles and Xie, 2023). In contrast, Large Language Models (LLMs) have demonstrated remarkable success using a two-stage training paradigm (Achiam et al., 2023; Dubey et al., 2024). After pretraining on vast and noisy web data, LLMs undergo a crucial second phase of fine-tuning on smaller but more specialized datasets. This two-stage approach allows the models to first develop broad capabilities and then align with user preferences to fulfill diverse human needs (Yang et al., 2024b; Dubey et al., 2024). The fine-tuning phase enhances model responsiveness to human expectations without significantly diminishing the broad capabilities acquired during pretraining. These alignment techniques have provided tremendous inspiration for the text-to-image domain, where leveraging human preference data for a given prompt holds great promise of unlocking diffusion models' ability to align with human expectations. However, diffusion models and LLMs operate under fundamentally different mechanisms—LLMs rely on autoregressive factorization while diffusion models are built with a chain of Markov transitions. As a result, porting these two-stage alignment methods to diffusion architectures remains a significant challenge.

In recent years, various RL-based methods have been developed for diffusion model alignment (Miao et al., 2024; Dai et al., 2023; Wallace et al., 2024; Li et al., 2024; Prabhudesai et al., 2023). Early-stage works focused on fine-tuning diffusion models through Reinforcement Learning from Human Feedback (RLHF) (Ouyang et al., 2022) after large-scale pretraining (Miao et al., 2024; Dai et al., 2023). These approaches typically first fit a reward model on human preference data, and then optimize the diffusion model to generate images that receive high reward scores while avoiding excessive deviation from the original model. However, building reliable reward models for diverse tasks

presents challenges, often requiring substantial datasets and considerable training resources (Wang et al., 2024; Wu et al., 2023; Xu et al., 2023). To address this issue, recent research has focused on alignment techniques without reliance on reward models. Diffusion-DPO (Wallace et al., 2024) was the first to avoid the reward model by extending the formulation of direct preference optimization (DPO) to diffusion models.

However, our analysis reveals that this method corresponds to minimizing the variational bound of the forward Kullback-Leibler (KL) divergence between the target and model distributions, which is known to encourage compromised mean-seeking behavior (Minka et al., 2005; Ji et al., 2024). As a result, the resulting model often tends to cover diverse human intent but fails to capture exact modes of human preference, resulting in blurry or diluted generations. Followup works (Li et al., 2024; Zhu et al., 2025) further propose learning with binary feedback instead of preference comparison, or fine-tuning the diffusion models from a score matching perspective. Nevertheless, none of these models challenge DPO's underlying formulation and thus the alignment performance degrades to a weak alignment method similar to supervised fine-tuning (SFT). As a result, developing principled and exact alignment methods for diffusion models remains an open problem. In this paper, we first provide the formal analysis that the Diffusion-DPO objective is equivalent to optimizing the variational upper bound of KL divergence between target and model distributions. Based on our observation, we revisit the diffusion alignment framework from the new distribution matching perspective and argue that, compared to minimizing forward KL, optimizing reverse KL divergence provides a more mode-seeking objective. Such an objective enables more precise optimization towards the major mode in target distribution, thereby more accurately capturing the true structure of human preferences. Based on this insight, we propose Divergence Minimization Preference Optimization (DMPO), a method that fine-tunes diffusion models by minimizing the reverse KL between the model distribution and the theoretical optimal distribution. Though sharing the same optimal solution as Diffusion-DPO, in the practical scenario with limited network expressivity, DMPO enables higher human preferences alignment behavior by pushing the probability mass to the main characteristics of the target distribution. Furthermore, we theoretically show that under mild assumptions, DMPO optimizes the policy distribution in the same direction as the original RL objective which provides justification for the rationality and effectiveness of our proposed algorithm.

Through extensive experiments on generation and editing tasks, we demonstrate that DMPO can consistently outperform current alignment methods for diffusion models in both automatic evaluation metrics and human evaluations, exhibiting precise and reliable preference alignment capabilities for generative diffusion models. The main contributions of this paper can be summarized as: (1) The first proposal of fine-tuning diffusion models through minimizing divergence based on human preferences, pioneering a new perspective for designing preference learning algorithms for diffusion models; (2) Theoretical analysis demonstrating that minimizing reverse KL achieves more precise alignment and the conditions and rationality for applying this to diffusion models; and (3) Experimental evidence showing that DMPO can more accurately understand prompts, enhance the diversity of generative models, and outperform existing preference learning baseline methods without any additional computational cost.

## 2 BACKGROUND

### 2.1 DIFFUSION MODELS

Suppose real data sample $\mathbf{x}_0$ follows data distribution $q(\mathbf{x}_0)$, denoising diffusion models (Ho et al., 2020) are generative models which have a discrete-time reverse process with a Markov structure $p_\theta(\mathbf{x}_{0:T}) = \prod_{t=1}^{T} p_\theta(\mathbf{x}_{t-1}|\mathbf{x}_t)$ where

$$p_\theta(\mathbf{x}_{t-1}|\mathbf{x}_t) = \mathcal{N}(x_{t-1}; \mu_\theta(\mathbf{x}_t), \sigma_{t|t-1}^2 \frac{\sigma_{t-1}^2}{\sigma_t^2} I). \tag{1}$$

Training the models is performed by minimizing the associated evidence lower bound (ELBO) (Kingma et al., 2021; Song et al., 2021a):

$$\mathcal{L}_{\text{Diff}} = \mathbb{E}_{\mathbf{x}_0, \boldsymbol{\epsilon}, t, \mathbf{x}_t} \left[ \omega(\lambda_t) \| \boldsymbol{\epsilon} - \boldsymbol{\epsilon}_\theta(\mathbf{x}_t, t) \|_2^2 \right], \tag{2}$$

with $\boldsymbol{\epsilon} \sim \mathcal{N}(0, \boldsymbol{I})$, $t \sim \mathcal{U}(0, T)$, $\mathbf{x}_t \sim q(\mathbf{x}_t|\mathbf{x}_0) = \mathcal{N}(\mathbf{x}_t; \alpha_t \mathbf{x}_0, \sigma_t^2 \boldsymbol{I})$. $\lambda_t = \alpha_t^2/\sigma_t^2$ is a signal-to-noise ratio (Kingma et al., 2021), and $\omega(\lambda_t)$ is a pre-specified weighting function (typically chosen to be constant (Ho et al., 2020; Song and Ermon, 2019)).

## 2.2 REINFORCEMENT LEARNING FROM HUMAN FEEDBACK (RLHF)

RLHF is a foundational paradigm for aligning large-scale generative models with human preferences. The standard RLHF pipeline consists of three stages: (1) supervised fine-tuning (SFT) on curated instruction data, (2) learning a reward model from human preference comparisons, and (3) reinforcement learning to fine-tune the policy using the learned reward. In the RL phase, a parameterized policy $p_\theta(\mathbf{x}_0|c)$ (input condition $c \sim \mathcal{D}_c$) is optimized to maximize expected reward under the guidance of a fixed reward function $r(c, \mathbf{x}_0)$, typically trained using a Bradley–Terry model (Bradley and Terry, 1952) on human-labeled preference pairs. To prevent the policy from drifting too far from a reference distribution $p_{\text{ref}}(\mathbf{x}_0|c)$ (often obtained from the SFT stage) the objective includes a KL-regularization term:

$$\max_{p_\theta} \mathbb{E}_{c\sim\mathcal{D}_c, \mathbf{x}_0\sim p_\theta(\mathbf{x}_0|c)} \left[r(c, \mathbf{x}_0)\right] - \beta \mathbb{D}_{\text{KL}} \left(p_\theta(\mathbf{x}_0|c) \,\|\, p_{\text{ref}}(\mathbf{x}_0|c)\right), \tag{3}$$

where $\beta > 0$ is a hyperparameter controlling regularization. Analytically, the optimal policy under this objective corresponds to a Boltzmann-rational distribution over the reference model (Peters and Schaal, 2007):

$$p^*(\mathbf{x}_0|c) = \frac{1}{Z(c)} p_{\text{ref}}(\mathbf{x}_0|c) \exp \left( \frac{1}{\beta} r(c, \mathbf{x}_0) \right), \tag{4}$$

where $Z(\mathbf{x})$ is the partition function. The goal of RLHF is to approximate this optimal policy $p^*$ using the trainable policy $p_\theta$.

**Direct Preference Optimization (DPO).** DPO bypasses the explicit reward learning and reinforcement learning steps in RLHF by directly modeling the optimal conditional distribution over outputs. Specifically, given preference pairs $(\mathbf{x}_0^w, \mathbf{x}_0^\ell)$ from $\mathcal{D}$, reward model can be rewrited as

$$r(c, \mathbf{x}_0) = \beta \log \frac{p_\theta(\mathbf{x}_0|c)}{p_{\text{ref}}(\mathbf{x}_0|c)} + \beta \log Z(c), \tag{5}$$

and substituting into the preference-based objective Equation (3) yields the DPO loss:

$$\mathcal{L}_{\text{DPO}}(\theta) = -\mathbb{E}_{c,\mathbf{x}_0^w,\mathbf{x}_0^\ell} \left[ \log \sigma \left( \beta \left( \log \frac{p_\theta(\mathbf{x}_0^w|c)}{p_{\text{ref}}(\mathbf{x}_0^w|c)} - \log \frac{p_\theta(\mathbf{x}_0^l|c)}{p_{\text{ref}}(\mathbf{x}_0^l|c)} \right) \right) \right] \tag{6}$$

This formulation allows the model to be trained directly via preference comparisons, without the need to estimate a reward model or compute policy gradients.

## 3 METHOD

### 3.1 REVISITING DIFFUSION-DPO

Diffusion-DPO is the first method to apply DPO to diffusion models. The approach proposes to minimize Equation (3) of diffusion models by replacing the KL-divergence term with its upper bound $\mathbb{D}_{\text{KL}} \left[ p_\theta(\mathbf{x}_{0:T}|c) \,\|\, p_{\text{ref}}(\mathbf{x}_{0:T}|c) \right]$, which is joint KL-divergence on sampling path $\mathbf{x}_{0:T}$. Based on the binary DPO formulation Equation (6), given text prompt $c$, image pairs $\mathbf{x}_0^w, \mathbf{x}_0^\ell$, Diffusion-DPO formulates the RLHF objective into the following objective:

$$\mathcal{L}_{\text{DPO-Diffusion}}(\theta) = -\mathbb{E}_{(\mathbf{x}_0^w,\mathbf{x}_0^\ell)\sim\mathcal{D}} \log \sigma \left( \beta \, \mathbb{E}_{\substack{\mathbf{x}_{1:T}^w \sim p_\theta(\mathbf{x}_{1:T}^w|\mathbf{x}_0^w) \\ \mathbf{x}_{1:T}^\ell \sim p_\theta(\mathbf{x}_{1:T}^\ell|\mathbf{x}_0^\ell)}} \left[ \log \frac{p_\theta(\mathbf{x}_{0:T}^w)}{p_{\text{ref}}(\mathbf{x}_{0:T}^w)} - \log \frac{p_\theta(\mathbf{x}_{0:T}^\ell)}{p_{\text{ref}}(\mathbf{x}_{0:T}^\ell)} \right] \right), \tag{7}$$

which allows the diffusion model to be optimized using pairwise preference data. By leveraging the convexity of the $-\log \sigma$ function and further simplifying the DPO objective across two denoising trajectories, such loss can be decomposed as per-step alignment loss. This enables the model to efficiently align with human preferences through direct optimization of the denoising process.

In this paper, we further revisit Diffusion-DPO from the distribution matching perspective. Interestingly, we obtain the following theoretical result:

**Theorem 1** *(informal) Generalizing Diffusion-DPO from the pairwise preference setting to the multi-sample setting with preference data sampled from the reference policy $p_{ref}$, we have that the gradient of Diffusion-DPO objective Equation* (7) *satisfies:*

$$\nabla_\theta \mathcal{L}_{\text{DPO-Diffusion}}(\theta) = \nabla_\theta \mathbb{E}_{\mathbf{x}\sim\mathcal{D}} \left[ \mathbb{D}_{\text{KL}} \left( \hat{p}^*(\mathbf{x}_{0:T}|c) \,\|\, \hat{p}_\theta(\mathbf{x}_{0:T}|c) \right) \right], \tag{8}$$

where $\hat{p}^*(\mathbf{x}_{0:T}|c) \propto p_{ref}(\mathbf{x}_{0:T}|c)\exp(r(\mathbf{x}_{0:T},c))$ and $\hat{p}_\theta(\mathbf{x}_{0:T}|c) \propto p_\theta(\mathbf{x}_{0:T}|c)^\beta p_{ref}(\mathbf{x}_{0:T}|c)^{1-\beta}$, i.e., Diffusion-DPO optimize the forward KL divergence $\mathbb{D}_{KL}\left(\hat{p}^*(\mathbf{x}_{0:T}|c) \parallel \hat{p}_\theta(\mathbf{x}_{0:T}|c)\right)$.

This states that Diffusion-DPO approximately minimizes a forward KL divergence between the whole sampling trajectory distribution of the preference-optimal policy and the learned policy. Extending this to the practical setting, Diffusion-DPO applies preference optimization on each step of the denoising process instead of the whole trajectory, which serves as an upper bound on the alignment objective. As a result, Diffusion-DPO similarly performs forward KL minimization to gradually align the model toward the optimal preference policy.

## 3.2 OUR METHOD

While DPO optimizes a forward KL divergence between the optimal policy and the learned policy, this formulation introduces a key limitation: it enforces a mean-seeking behavior. Specifically, minimizing the Forward KL penalizes the learned policy $p_\theta$ harshly whenever it assigns low probability to any output favored by the optimal policy $p^*$. This encourages $p_\theta$ to cover the full support of $p^*$, often resulting in diffuse generations that fail to concentrate on highly preferred samples. In contrast, minimizing the reverse KL divergence provides a more targeted form of alignment. The reverse KL is given by $\mathbb{D}_{KL}\left(\hat{p}_\theta(\mathbf{x}_{0:T}|c) \parallel \hat{p}^*(\mathbf{x}_{0:T}|c)\right)$, and only penalizes the model when it places mass in regions unsupported by $p^*$. This drives the model toward the high-reward modes of $p^*$, promoting sharper and more precise alignment with human preferences. Figure 1 provides a toy 2D illustration of this contrast, where forward KL exhibits mean-seeking behavior while reverse KL concentrates on the dominant mode. A more detailed analysis is provided in Section 5.1.

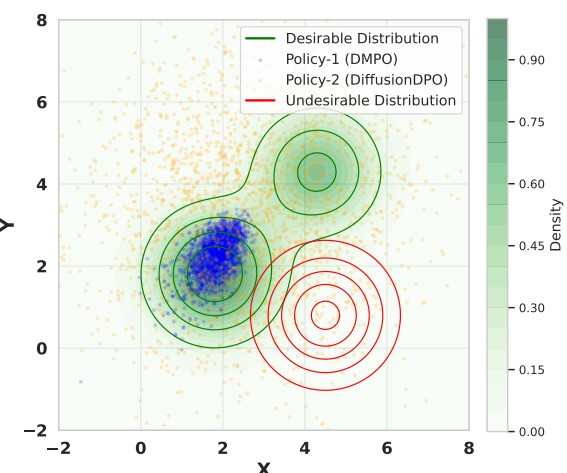

Figure 1: Visualization of the effect of forward KL (DiffusionDPO) and reverse KL (DMPO) alignment on policy learning. We sample from MLP diffusion models trained with DPO and DMPO objectives. Green contours indicate the desirable distribution, while red contours denote the undesirable distribution. Compared to DiffusionDPO (orange samples), DMPO (blue samples) aligns more accurately with the target by concentrating on the dominant mode of the mixture Gaussian, highlighting its stronger alignment capability.

Based on insight, we propose *Divergence Minimization Preference Optimization* (*DMPO*), which replaces the *forward KL* with a *reverse KL* objective to align diffusion models with human preferences by minimizing a divergence between the learned policy and the latent optimal policy derived from Equation (4). According to the optimal policy $p^*$ in Equation (4), we can rewrite the reward function in the form of $r(c, \mathbf{x}_{0:T}) = \log \hat{p}^*(\mathbf{x}_{0:T}|c) - \log p_{ref}(\mathbf{x}_{0:T}|c)$. Besides, we can define the relative logits $g_\theta(c, \mathbf{x}_{0:T}) = \log \hat{p}_\theta(\mathbf{x}_{0:T}|c) - \log p_{ref}(\mathbf{x}_{0:T}|c) = \beta(\log p_\theta(\mathbf{x}_{0:T}) - \log p_{ref}(\mathbf{x}_{0:T}))$. Putting $r(c, \mathbf{x}_{0:T})$ and $g_\theta(c, \mathbf{x}_{0:T})$ into $\mathbb{D}_{KL}\left(\hat{p}_\theta(\mathbf{x}_{0:T}|c) \parallel \hat{p}^*(\mathbf{x}_{0:T}|c)\right)$, the divergence objective becomes:

$$\mathcal{L}(\theta) = \mathbb{D}_{KL}\left(\hat{p}_\theta(\mathbf{x}_{0:T}|c) \parallel \hat{p}^*(\mathbf{x}_{0:T}|c)\right) = \mathbb{E}_{\hat{p}_\theta}\left[\log\left(\frac{\hat{p}_\theta}{p_{ref}} \cdot \frac{p_{ref}}{\hat{p}_*}\right)\right] = \mathbb{E}_{p_{ref}}\left[p^{g_\theta}\log\left(\frac{p^{g_\theta}}{p^r}\right)\right], \quad (9)$$

where $p^{g_\theta}$ and $p^r$ are unnormalized distributions proportional to $\exp(g_\theta)$ and $\exp(r)$ over $K$ samples respectively. Formly they can be written as :

$$p^{g_\theta}\left(i \mid \{\mathbf{x}_{0:T}\}_{1:K}, c\right) = \frac{e^{g_\theta(c,\{\mathbf{x}_{0:T}\}_i)}}{\sum_{j=1}^K e^{g_\theta(c,\{\mathbf{x}_{0:T}\}_j)}}, \quad p^r\left(i \mid \{\mathbf{x}_{0:T}\}_{1:K}, c\right) = \frac{e^{r(c,\{\mathbf{x}_{0:T}\}_i)}}{\sum_{j=1}^K e^{r(c,\{\mathbf{x}_{0:T}\}_j)}}. \quad (10)$$

**Learning with Pair-wise Preference Data**. We consider the practical objective given a dataset $\mathcal{D}$ consisting of text prompts $c$ and preference-labeled output pairs $(\mathbf{x}_0^w, \mathbf{x}_0^\ell)$, where $\mathbf{x}_0^w$ is preferred over

$\mathbf{x}_0^\ell$. By setting $K = 2$ in Equation (10), Equation (9) can be expressed as:

$$\mathcal{L}(\theta) = \mathbb{E}_{(\mathbf{x}_0^w, \mathbf{x}_0^l) \sim \mathcal{D}} \left[ \sigma(u(\theta)) \log \frac{\sigma(u(\theta))}{1 - \alpha} + \sigma(-u(\theta)) \log \frac{\sigma(-u(\theta))}{\alpha} \right], \quad (11)$$

where $u(\theta) = g_\theta(c, \mathbf{x}_{0:T}^w) - g_\theta(c, \mathbf{x}_{0:T}^\ell) = \beta \cdot \mathbb{E}_{\substack{\mathbf{x}_{1:T}^w \sim p_\theta(\mathbf{x}_{1:T}^w | \mathbf{x}_0^w) \\ \mathbf{x}_{1:T}^l \sim p_\theta(\mathbf{x}_{1:T}^l | \mathbf{x}_0^l)}} \left[ \log \frac{p_\theta(\mathbf{x}_{0:T}^w)}{p_{\text{ref}}(\mathbf{x}_{0:T}^w)} - \log \frac{p_\theta(\mathbf{x}_{0:T}^l)}{p_{\text{ref}}(\mathbf{x}_{0:T}^l)} \right].$

Note we omit $c$ as it's always independent of the path$\mathbf{x}_{0:T}$ as it does not affect the derivation (more details are included in Appendix A).

Since $u(\theta)$ is an expectation over the trajectory $\mathbf{x}_{1:T}$, it is computationally intractable. To address this, we instead aim to estimate an upper bound of the original objective by applying Jensen's inequality. To this end, we analyze the convexity of the function $f(u)$ from Equation (11):

$$f(u) = \sigma(u) \log \frac{\sigma(u)}{1-\alpha} + \sigma(-u) \log \frac{\sigma(-u)}{\alpha}. \quad (12)$$

By analyzing the second derivative $f''(u)$, we observe that for small $\alpha > 0$, there always exists an interval $(h_1(\alpha), h_2(\alpha))$ where $f(u)$ is strictly convex. In practice, the model is initialized near $u = 0$ and optimization drives $u$ positively as it emphasizes the winner distribution. The optimum occurs at $u^* = \log \frac{1-\alpha}{\alpha}$, and for any $\alpha \in (0, 1)$, we verify that $h_2(\alpha) > u^*$, ensuring that the practical optimization trajectory $u \in (0, u^*)$ remains within the convexity-preserving region.

This analysis relies on the mild assumption that $(h_1(\alpha), h_2(\alpha))$ is sufficiently wide for alignment. A detailed derivation of this interval as a function of $\alpha$ is provided in Appendix A. Under this assumption we can use the Jensen inequality to move the expectation of $\mathbf{x}$ in Equation (11) outside:

$$\mathcal{L}(\theta) \le \mathbb{E}_{\substack{(\mathbf{x}_0^w, \mathbf{x}_0^l) \sim \mathcal{D}, t \sim \mathcal{U}(0, T) \\ \mathbf{x}_t^w \sim p_\theta(\mathbf{x}_{t-1}^w, \mathbf{x}_t^w | \mathbf{x}_0^w) \\ \mathbf{x}_t^l \sim p_\theta(\mathbf{x}_{t-1}^l, \mathbf{x}_t^l | \mathbf{x}_0^l)}} \left[ \sigma(u_t(\theta)) \log \frac{\sigma(u_t(\theta))}{1 - \alpha} + \sigma(-u_t(\theta)) \log \frac{\sigma(-u_t(\theta))}{\alpha} \right], \quad (13)$$

where $u_t(\theta) = \beta T \left( \log \frac{p_\theta(\mathbf{x}_{t-1}^w | \mathbf{x}_t^w)}{p_{\text{ref}}(\mathbf{x}_{t-1}^w | \mathbf{x}_t^w)} - \log \frac{p_\theta(\mathbf{x}_{t-1}^l | \mathbf{x}_t^l)}{p_{\text{ref}}(\mathbf{x}_{t-1}^l | \mathbf{x}_t^l)} \right)$. In order to avoid intractable true posterior and enable efficient training, we approximate the reverse process $p_\theta(\mathbf{x}_{t-1}, \mathbf{x}_t | \mathbf{x}_0)$ using the forward distribution $q(\mathbf{x}_{1:T} | \mathbf{x}_0)$, which leads to the final loss:

$$\mathcal{L}_{\text{DMPO}}(\theta) = \mathbb{E}_{\substack{(\mathbf{x}_0^w, \mathbf{x}_0^l) \sim \mathcal{D}, t \sim \mathcal{U}(0, T) \\ \mathbf{x}_t^w \sim q(\mathbf{x}_t^w | \mathbf{x}_0^w) \\ \mathbf{x}_t^l \sim q(\mathbf{x}_t^l | \mathbf{x}_0^l)}} \left[ \sigma(u_t(\theta)) \log \frac{\sigma(u_t(\theta))}{1 - \alpha} + \sigma(-u_t(\theta)) \log \frac{\sigma(-u_t(\theta))}{\alpha} \right], \quad (14)$$

where $u_t(\theta)$ is further simplified into the following form:

$$u_t(\theta) = -\beta T \Big( \mathbb{D}_{\text{KL}}(q(\mathbf{x}_{t-1}^w | \mathbf{x}_0^w, \mathbf{x}_t^w) \,\|\, p_\theta(\mathbf{x}_{t-1}^w | \mathbf{x}_t^w)) - \mathbb{D}_{\text{KL}}(q(\mathbf{x}_{t-1}^w | \mathbf{x}_0^w, \mathbf{x}_t^w) \,\|\, p_{\text{ref}}(\mathbf{x}_{t-1}^w | \mathbf{x}_t^w))$$

$$- \mathbb{D}_{\text{KL}}(q(\mathbf{x}_{t-1}^l | \mathbf{x}_0^l, \mathbf{x}_t^l) \,\|\, p_\theta(\mathbf{x}_{t-1}^l | \mathbf{x}_t^l)) + \mathbb{D}_{\text{KL}}(q(\mathbf{x}_{t-1}^l | \mathbf{x}_0^l, \mathbf{x}_t^l) \,\|\, p_{\text{ref}}(\mathbf{x}_{t-1}^l | \mathbf{x}_t^l)) \Big)$$

$$= -\beta T \omega(\lambda_t) \Big( (\|\epsilon^w - \epsilon_\theta(\mathbf{x}_t^w, t)\|_2^2 - \|\epsilon^w - \epsilon_{\text{ref}}(\mathbf{x}_t^w, t)\|_2^2) - (\|\epsilon^l - \epsilon_\theta(\mathbf{x}_t^l, t)\|_2^2 - \|\epsilon^l - \epsilon_{\text{ref}}(\mathbf{x}_t^l, t)\|_2^2) \Big)$$

with $\epsilon \sim \mathcal{N}(0, \mathbf{I})$, $t \sim \mathcal{U}(0, T)$, $\mathbf{x}_t \sim q(\mathbf{x}_t | \mathbf{x}_0) = \mathcal{N}(\mathbf{x}_t; \alpha_t \mathbf{x}_0, \sigma_t^2 \mathbf{I})$. $\lambda_t = \alpha_t^2 / \sigma_t^2$ is a signal-to-noise ratio, $\omega(\lambda_t)$ is a pre-specified weighting function). $T$ is the maximum of training timesteps, we merge $T$ and $\omega(\lambda_t)$ into $\beta$ finally.

### 3.3 THEORETICAL ANALYSIS

In this section, we further provide the theoretical analysis of the DMPO method by showing its connection with the original RLHF objective. Specifically, we show that the optimization direction of DMPO aligns with that of maximizing the RLHF objective in Equation (3). Let the RLHF loss for diffusion path $\mathbf{x}_{0:T}$ be denoted as $\mathcal{L}_{\text{RLHF}}$:

$$\mathcal{L}_{\text{RLHF}}(\theta) = \mathbb{E}_{c \sim \mathcal{D}_c, \mathbf{x}_{0:T} \sim p_\theta(\mathbf{x}_{0:T} | c)} [r(c, \mathbf{x}_0)] - \beta \mathbb{D}_{\text{KL}} [p_\theta(\mathbf{x}_{0:T} | c) \,\|\, p_{\text{ref}}(\mathbf{x}_{0:T} | c)]. \quad (15)$$

Starting from the divergence-based formulation in Equation (9), we formally have the following connection of the DMPO objective and the RL method objective.

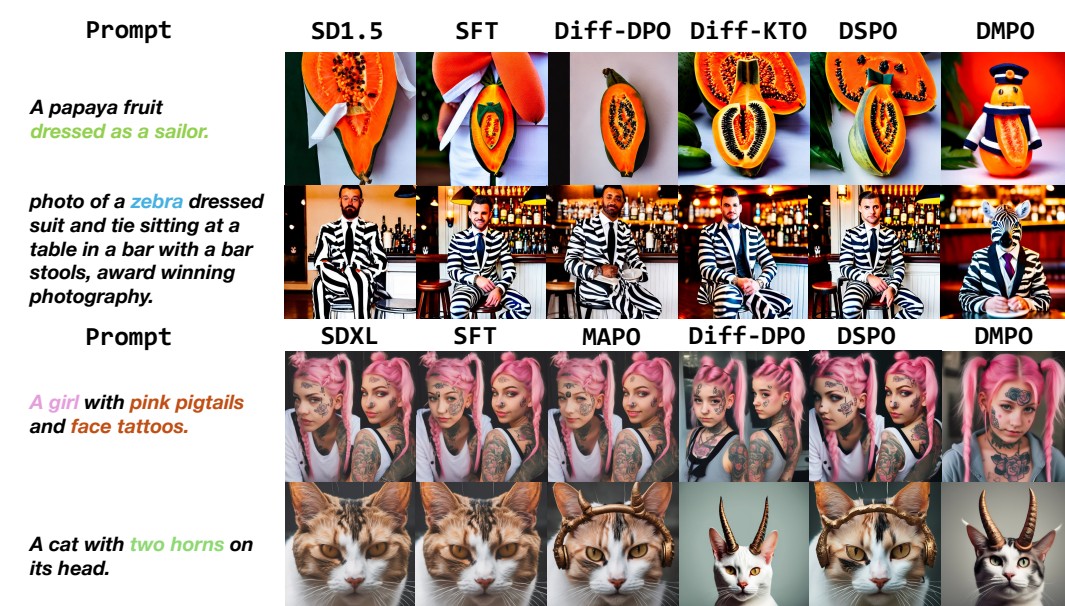

Figure 2: **Qualitative result of different alignment methods**. We show the images generated by different models for various prompts which are selected from Pick-a-Pic V2, Parti-Prompt and HPS V2. The top two rows present results based on SD1.5, while the bottom two rows are based on SDXL. "Diff" represents "Diffusion" for simplicity.

**Theorem 2** *Generalizing* DMPO *from the pairwise preference setting to the multi-sample setting with preference data sampled from the reference policy $p_{ref}$, when $\beta = 1$, we will have that the gradient of the* DMPO *objective satisfies:*

$$\nabla_\theta \mathcal{L}_{\text{DMPO}}(\theta) = \nabla_\theta \mathbb{E}_{c \sim \mathcal{D}_c, \mathbf{x}_{0:T} \sim p_\theta(\mathbf{x}_{0:T}|c)} \left[ \mathbb{D}_{\text{KL}}(\hat{p}_\theta(\mathbf{x}_{0:T}|c) \| \hat{p}^*(\mathbf{x}_{0:T}|c)) \right] = -\nabla_\theta \mathcal{L}_{RLHF}(\theta). \tag{16}$$

Theorem 2 states that the optimization direction for $p_\theta(\mathbf{x}_{0:T})$ aligns with that of RLHF, which theoretically justifies that the objective of our DMPO is consistent with reinforcement learning–based alignment under specific conditions. The full technical derivation is provided in Appendix A. This highlights that though derived from a divergence minimization view, DMPO is consistent with the optimization direction of the original RLHF method.

## 4 RELATED WORK

Diffusion model alignment seeks to steer generative outputs toward human preferences by embedding reinforcement learning objectives within the denoising process (Yang et al., 2024a; Xu et al., 2023; Miao et al., 2024). Early efforts such as DDPO (Black et al., 2024) leverage task-specific, hand-crafted reward functions—e.g., promoting compressibility—to fine-tune pretrained diffusion backbones. More recently, DPOK (Fan et al., 2023) replaces these bespoke rewards with feedback signals distilled from AI agents trained on vast human-preference corpora. Diffusion-DPO (Wallace et al., 2024) adapts the core DPO framework, directly ingesting pairwise preference data to update the diffusion model. Diffusion-KTO (Li et al., 2024) extends utility-based preference optimization to diffusion models using only binary human feedback and DSPO (Zhu et al., 2025). DSPO derives a novel step-wise alignment method for diffusion models from the perspective of score-based modeling. However, these approaches cannot precisely align the learned policy with the optimal policy derived from RLHF objective. In this work, we instead approach alignment by minimizing the divergence between the learned policy and the optimal policy, and adopt a reverse KL objective to explicitly align them. Efficient Exact Optimization (EXO) (Ji et al., 2024) and $f$-divergence Preference Optimization ($f$-PO) (Han et al., 2024) also explored aligning language models (LMs) using reverse KL. However, all these studies are limited to autoregressive language models, while our method focuses on diffusion models and provides fundamentally different inspiration for aligning the Markov chains.

# 5 EXPERIMENTS

## 5.1 ILLUSTRATIVE STUDY OF ALIGNMENT OBJECTIVES

To further illustrate the behavior of different alignment objectives, we design a toy 2D experiment. The pretraining data is sampled from a Gaussian distribution with mean $(3, 3)$ and variance 2. Preference data comes from a mixture of two Gaussians with means $(4.3, 4.3)$ and $(1.8, 1.8)$ (variance 1, ratio 6:4). Undesirable data is drawn from a Gaussian with mean $(4.5, 0.8)$ and variance 1. We first pretrain a small MLP diffusion model on the pretraining data using the standard diffusion objective, and then fine-tune it with different alignment objectives. From each trained model, we sample 2000 points. Figure 1 shows the results. DiffusionDPO pushes samples away from the undesirable region but distributes them rather uniformly, exhibiting mean-seeking behavior. In contrast, DMPO captures the dominant mode of the mixture distribution, demonstrating mode-seeking behavior.

## 5.2 EXPERIMENTAL SETTING

**Dataset and Baselines**. We select Stable Diffusion v1.5 (SD1.5) and Stable Diffusion XL Base 1.0 (SDXL) as our base models. We train DMPO on the Pick-a-Pic V2 (Kirstain et al., 2023) dataset, which consists of pairwise preferences for images generated by SDXL-beta and Dreamlike (a fine-tuned version of SD1.5). After excluding the $\sim 12\%$ of pairs with ties, we end up with 851,293 preference pairs across 58,960 unique prompts. Our baselines include diffusion models fine-tuned on the Pick-a-Pic V2 dataset using various alignment methods, based on either SD1.5 or SDXL: Diffusion-DPO (Wallace et al., 2024), Diffusion-KTO (Li et al., 2024), DSPO (Zhu et al., 2025), MAPO (Hong et al., 2025). We also compare against the original SD1.5 and SDXL model as well as a supervised fine-tuning (SFT) SD1.5 model. We implement SFT and train DSPO by their released repo ourselves, and for Diffusion-DPO, KTO, and MAPO, we use the officially released checkpoints. In addition, we further compare against Stable Diffusion 3.5 Medium (SD3.5-M) (Esser et al., 2024) models fine-tuned with DPO and with our proposed DMPO objective to assess the effectiveness of these methods on a stronger backbone.

**Training Details**. We detail the implementation setup for DMPO in this section. Training is distributed across 4 NVIDIA A100 40GB GPUs, each processing a local batch of 2 pairs, with gradient accumulation over 256 steps to achieve the desired global batch size 2048. All models are trained at fixed square resolutions using a learning rate of $1 \times 10^{-5}$, with a linear warmup spanning the first 500 steps. We select the best-performing model from training for evaluation; for DPMO with SD1.5 as the base model, we set the smoothing coefficient $\alpha = 0.01$ and $\beta = 2000$, for DPMO with SDXL as the base model, we use $\alpha = 0.01$ and $\beta = 6000$, and for DPMO with SD3.5-M as the base model, we use $\alpha = 0.1$ and $\beta = 100$. For more details on the model's behavior under various parameter settings, please refer to Appendix C.4.

**Evaluation Details**. We evaluate the performance of DPMO through both automated preference metrics and human user studies. For **general alignment evaluation**, we choose test prompts from Pick-a-Pic V2, HPS V2, and PartiPrompt, and compare DMPO against all baselines on text-to-image generation task. For **image editing evalution**, we evaluate editing performance on two standard benchmarks, TEd-bench (Kawar et al., 2023) and InstructPix2Pix (Brooks et al., 2023) Bench by using SDEdit (Meng et al., 2022) pipeline and compare DMPO against all baselines. To ensure a fair comparison, we ensure consistency across all models, i.e., setting the guidance scale to 7.5, and the number of sampling steps to 50. For automated evaluation, we report three aspects: (1) Reward Score directly output by the reward models, (2) Win rates comparing DPMO and all baselines against the original base model, and (3) Pairwise win rates comparing DPMO directly with each baseline. The preference metrics include: **PickScore** (Kirstain et al., 2023), **HPS V2** (Wu et al., 2023), **CLIP** (Radford et al., 2021), **LAION Aesthetics Classifier** (Schuhmann, 2022), and **ImageReward** (Xu et al., 2023), all of which are caption-aware models trained to predict human preference scores based on an image and its associated prompt. In addition, we conduct user study to compare DPMO with existing baselines. Each comparison is assessed from three perspectives: *Q1. General Preference*: Which image do you prefer given the prompt? *Q2. Visual Appeal*: Which image is more visually appealing regardless of the prompt? and *Q3. Prompt Alignment*: Which image better matches the text description?. We sampled 100 test prompts from Pick-a-Pic V2, HPS V2, and PartiPrompt, and evaluated DMPO (based on SD1.5), SD1.5, and all baselines. More details on the experimental setup is provided in the Appendix C.

Table 1: Reward Score comparisons on Pick-a-Pic V2, HPS V2 and Parti-Prompt datasets for all baselines versus SD1.5, best results are in **boldface**. For simplicity, "Diff" represents "Diffusion".

| Dataset | Method | Pick Score(↑) | HPS(↑) | CLIP(↑) | Aesthetics (↑) | Image Reward (↑) |
|---|---|---|---|---|---|---|
| Pick-a-Pic V2 | SD1.5 | 0.2066 | 0.2612 | 0.3254 | 5.3200 | -0.1478 |
| | DMPO | **0.2165** | **0.2705** | **0.3453** | **5.6304** | **0.5415** |
| | SFT | 0.2124 | 0.2701 | 0.3431 | 5.5213 | 0.4953 |
| | Diff-DPO | 0.2106 | 0.2642 | 0.3337 | 5.4729 | 0.0750 |
| | Diff-KTO | 0.2120 | 0.2701 | 0.3383 | 5.5848 | 0.5341 |
| | DSPO | 0.2120 | 0.2696 | 0.3396 | 5.6022 | 0.4683 |
| HPS V2 | SD1.5 | 0.2089 | 0.2672 | 0.3458 | 5.4249 | -0.1175 |
| | DMPO | **0.2195** | 0.2768 | **0.3629** | **5.7997** | **0.6350** |
| | SFT | 0.2159 | 0.2771 | 0.3537 | 5.7356 | 0.5607 |
| | Diff-DPO | 0.2133 | 0.2706 | 0.3529 | 5.6014 | 0.1271 |
| | Diff-KTO | 0.2154 | **0.2775** | 0.3547 | 5.7381 | 0.5652 |
| | DSPO | 0.2155 | 0.2772 | 0.3551 | 5.7483 | 0.5385 |
| Parti Prompt | SD1.5 | 0.2139 | 0.2679 | 0.3322 | 5.3115 | 0.0196 |
| | DMPO | **0.2205** | **0.2758** | **0.3483** | **5.5438** | **0.6614** |
| | SFT | 0.2172 | 0.2751 | 0.3401 | 5.5277 | 0.5293 |
| | Diff-DPO | 0.2163 | 0.2700 | 0.3382 | 5.3866 | 0.2243 |
| | Diff-KTO | 0.2173 | **0.2758** | 0.3408 | 5.5098 | 0.5551 |
| | DSPO | 0.2174 | 0.2755 | 0.3382 | 5.5291 | 0.5021 |

Table 2: Reward Score comparisons on Pick-a-Pic V2, HPS V2 and Parti-Prompt datasets for all baselines versus SDXL, best results are in **boldface**. For simplicity, "Diff" represents "Diffusion".

| Dataset | Method | Pick Score(↑) | HPS(↑) | CLIP(↑) | Aesthetics (↑) | Image Reward (↑) |
|---|---|---|---|---|---|---|
| Pick-a-Pic V2 | SDXL | 0.2203 | 0.2661 | 0.3609 | 5.9892 | 0.5111 |
| | DMPO | **0.2264** | **0.2730** | **0.3741** | 5.9422 | **0.8563** |
| | SFT | 0.2224 | 0.2675 | 0.3624 | 5.9239 | 0.5834 |
| | Diff-DPO | 0.2256 | 0.2709 | 0.3722 | 5.9890 | 0.7584 |
| | MAPO | 0.2213 | 0.2682 | 0.3615 | **6.1196** | 0.6226 |
| | DSPO | 0.2256 | 0.2684 | 0.3615 | 5.9598 | 0.6831 |
| HPS V2 | SDXL | 0.2271 | 0.2730 | 0.3775 | 6.1125 | 0.6749 |
| | DMPO | **0.2320** | **0.2804** | **0.3914** | 6.1883 | **1.0169** |
| | SFT | 0.2277 | 0.2762 | 0.3784 | 6.0638 | 0.6998 |
| | Diff-DPO | 0.2314 | 0.2777 | 0.3890 | 6.1546 | 0.9610 |
| | MAPO | 0.2279 | 0.2765 | 0.3801 | **6.2134** | 0.7839 |
| | DSPO | 0.2285 | 0.2754 | 0.3795 | 6.0545 | 0.7385 |
| Parti Prompt | SDXL | 0.2249 | 0.2714 | 0.3551 | 5.7648 | 0.6010 |
| | DMPO | **0.2290** | **0.2780** | **0.3769** | 5.8598 | **1.0399** |
| | SFT | 0.2257 | 0.2701 | 0.3531 | 5.7239 | 0.6953 |
| | Diff-DPO | 0.2286 | 0.2763 | 0.3688 | 5.7900 | 0.9515 |
| | MAPO | 0.2250 | 0.2732 | 0.3561 | **5.9089** | 0.7031 |
| | DSPO | 0.2256 | 0.2714 | 0.3596 | 5.7598 | 0.7600 |

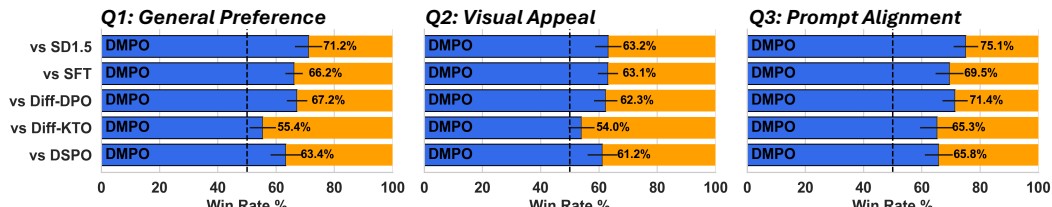

Figure 3: **User Study Results**. DMPO significantly outperforms all baselines in human evaluation across three evaluation questions.

### 5.3 ALIGNMENT RESULT

**Qualitative Result**. Figure 2 presents a qualitative comparison of generations from DMPO and other baseline models under the same prompt and control conditions. Compared to alternative methods, DMPO consistently demonstrates a stronger ability to capture the semantic intent of the prompt, producing outputs that are both more accurate and higher in quality. For instance, in the second row, it is the only model that correctly depicts "the papaya dressed as a sailor". Additional qualitative results can be found in the Appendix C.

Table 3: Reward Score comparisons on Pick-a-Pic V2, HPS V2 and Parti-Prompt datasets for all baselines versus SD3.5-M, best results are in **boldface**.

| Dataset | Method | Pick Score(↑) | HPS(↑) | CLIP(↑) | Aesthetics (↑) | Image Reward (↑) |
|---------|--------|---------------|--------|---------|----------------|------------------|
| Pick-a-Pic V2 | SD3.5-M | 0.2223 | 0.2848 | 0.3579 | 5.8989 | 0.9789 |
| | SD3.5-DPO | 0.2281 | 0.2890 | 0.3631 | 6.0040 | 1.2241 |
| | SD3.5-DMPO | **0.2294** | **0.2904** | **0.3641** | **6.0876** | **1.2456** |
| HPS V2 | SD3.5-M | 0.2289 | 0.2917 | 0.3733 | 6.0018 | 1.1211 |
| | SD3.5-DPO | 0.2342 | 0.2948 | 0.3722 | 6.1103 | 1.2775 |
| | SD3.5-DMPO | **0.2373** | **0.2984** | **0.3762** | **6.1883** | **1.2874** |
| Parti Prompt | SD3.5-M | 0.2283 | 0.2915 | 0.3620 | 5.6392 | 1.1572 |
| | SD3.5-DPO | 0.2349 | **0.2985** | 0.3781 | 5.8215 | 1.4443 |
| | SD3.5-DMPO | **0.2368** | 0.2983 | **0.3799** | **5.8598** | **1.4632** |

Figure 4: Images edited by different models for various prompts which are selected from TEd-bench. DMPO significantly outperforms other baselines in both text alignment and visual quality.

**Quantitative Result**. Tables 1 to 3 report the reward-model scores for DMPO across different base models and test sets. We find that, for every reward metric, the DMPO-tuned models consistently surpass their corresponding base models. Notably, in terms of PickScore, DMPO achieves the best results across all base models and all test sets. Moreover, Tables 4 to 6 show the win rate comparisons: while DMPO generally outperforms nearly all baselines, it is particularly strong in the SD1.5-based setting, where DMPO based on SD1.5 exceeds all baseline models by at least 64.6% on every dataset. further highlighting the effectiveness of the proposed divergence minimization objective.

## 5.4 IMAGE EDITING RESULTS

Beyond improving alignment in image generation tasks, DMPO also significantly enhances the model's capabilities in image editing, particularly for text-guided image editing scenarios. This improvement stems from the model's strengthened ability to interpret and execute complex textual instructions. Figure 4 presents representative qualitative editing results on TEd-bench. In the first rows, only DMPO correctly understands and represents content 'hugging'. In the second row, where the input prompt is "A teddy bear holding a cup.", only DMPO generates an image that is both semantically faithful and highly visually appealing. More results can be seen in Appendix C.

## 6 CONCLUSION

In this work, we propose *Divergence Minimization Preference Optimization (*DMPO*)*, a novel and theoretically grounded framework for aligning diffusion models with human preferences. Unlike prior methods that rely on forward KL divergence or implicit reward modeling, DMPO leverages reverse KL divergence to achieve mode-seeking behavior, enabling more precise and robust preference alignment. We provide a rigorous derivation of DMPO under the diffusion framework, including the formulation of alignment intervals. Through comprehensive experiments across multiple datasets and reward models, we demonstrate that DMPO consistently outperforms existing alignment baselines, showing stronger adherence to user preference. DMPO offers a principled and effective solution to the challenge of preference alignment in diffusion models, bridging theoretical insights and practical performance. We hope this work inspires further research into divergence-based alignment objectives for text-to-image generative models.

## ETHICS STATEMENT

All authors have read and agreed to comply with the ICLR Code of Ethics (https://iclr.cc/public/CodeOfEthics). The datasets used in this work are publicly available without including any personally identifiable or sensitive information. All datasets comply with their original licenses.

## REPRODUCIBILITY STATEMENT

We are committed to ensuring the reproducibility of our results. The methodology and evaluation setup are fully documented in the main paper and appendix. To further support replication and extension of our work, we release the implementation code to enable other researchers to replicate and extend our findings.

**Code Availability** All code is available at an anonymous GitHub repository, which can reproduces the experiments in the paper: https://anonymous.4open.science/r/dmpo-iclr2026.

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

## USE OF LARGE LANGUAGE MODELS

We used large language models (LLMs) only for minor language editing and grammar improvement of the manuscript. No LLMs were used for research ideation, experimental design, analysis, or writing of technical content.

## OVERVIEW

Here, we provide an overview of the Appendix below:

- §A presents theoretical analysis about all the theorems and our method.
- §B details the evaluation and includes pseudo-code for reproducibility.
- §C provide more quantitative and qualitative results for DMPO fine-tuned on different base models.

## A  PROOFS AND DERIVATIONS

### A.1  PROOF OF THEOREM 1

In this section, we provide a detailed proof of Theorem 1, demonstrating that the objective of DPO corresponds to the forward KL divergence. Starting from Equation (10) and Equation (9), we extand the dpo loss $\mathcal{L}_{dpo}(K=2)$ from pair preference data to K samples selection, where the sigmoid needs to be replaced with softmax to present the soft distribution. Utilizing the definition of $\hat{p}_\theta(\mathbf{x}_{0:T})$ :

$$\frac{\hat{p}_\theta(\mathbf{x}_{0:T}|\mathbf{c})}{p_{ref}(\mathbf{x}_{0:T}|\mathbf{c})} \propto \left( \frac{p_\theta(\mathbf{x}_{0:T}|\mathbf{c})}{p_{ref}(\mathbf{x}_{0:T}|\mathbf{c})} \right)^\beta$$

and by combining Equation (5), we can extend the dpo loss to:

$$\mathcal{L}_{\mathrm{dpo}}(\theta) = \mathbb{E}_{c\sim\mathcal{D}_c}\mathbb{E}_{p_{\mathrm{ref}}(\{\mathbf{x}_{0:T}\}_{1:K}|c)} \left[ -\sum_{i=1}^{K} \frac{e^{r(c,\{\mathbf{x}_{0:T}\}_i)}}{\sum_{j=1}^{K} e^{r(c,\{\mathbf{x}_{0:T}\}_j)}} \log \left( \frac{e^{\beta \log \frac{p_\theta(\{\mathbf{x}_{0:T}\}_i|c)}{p_{\mathrm{ref}}(\{\mathbf{x}_{0:T}\}_i|c)}}}{\sum_{j=1}^{K} e^{\beta \log \frac{p_\theta(\{\mathbf{x}_{0:T}\}_j|c)}{p_{\mathrm{ref}}(\{\mathbf{x}_{0:T}\}_j|c)}}} \right) \right]$$

$$= \mathbb{E}_{c\sim\mathcal{D}_c}\mathbb{E}_{p_{\mathrm{ref}}(\{\mathbf{x}_{0:T}\}_{1:K}|c)} \left[ -\sum_{i=1}^{K} \frac{e^{r(c,\{\mathbf{x}_{0:T}\}_i)}}{\sum_{j=1}^{K} e^{r(c,\{\mathbf{x}_{0:T}\}_j)}} \log \left( \frac{\log \frac{p_\theta(\{\mathbf{x}_{0:T}\}_i|c)}{p_{\mathrm{ref}}(\{\mathbf{x}_{0:T}\}_i|c)}^{\beta}}{\sum_{j=1}^{K} \log \frac{p_\theta(\{\mathbf{x}_{0:T}\}_j|c)}{p_{\mathrm{ref}}(\{\mathbf{x}_{0:T}\}_j|c)}^{\beta}} \right) \right]$$

$$= \mathbb{E}_{c\sim\mathcal{D}_c}\mathbb{E}_{p_{\mathrm{ref}}(\{\mathbf{x}_{0:T}\}_{1:K}|c)} \left[ -\sum_{i=1}^{K} \frac{e^{r(c,\{\mathbf{x}_{0:T}\}_i)}}{\sum_{j=1}^{K} e^{r(c,\{\mathbf{x}_{0:T}\}_j)}} \log \left( \frac{\frac{\hat{p}_\theta(\{\mathbf{x}_{0:T}\}_i|c)}{p_{\mathrm{ref}}(\{\mathbf{x}_{0:T}\}_i|c)}}{\sum_{j=1}^{K} \frac{\hat{p}_\theta(\{\mathbf{x}_{0:T}\}_j|c)}{p_{\mathrm{ref}}(\{\mathbf{x}_{0:T}\}_j|c)}} \right) \right] \quad (17)$$

When $K \to \infty$, for arbitrary function $g$, the estimate $\frac{1}{K}\sum_{i=1}^{K} g(\{\mathbf{x}_{0:T}\}_i)$ is unbiased since $\{\{\mathbf{x}_{0:T}\}_i\}_{i=1}^{K}$ are sampled from $p_{\mathrm{ref}}(\cdot|c)$,, i.e.,

$$\lim_{K\to\infty} \frac{1}{K}\sum_{i=1}^{K} g(\{\mathbf{x}_{0:T}\}_i) = \mathbb{E}_{p_{\mathrm{ref}}(\mathbf{x}_{0:T}|c)}[g(\mathbf{x}_{0:T})].$$

For $g$ is the relative $g(c, \mathbf{x}_{0:T}) = \frac{\hat{p}_\theta(\mathbf{x}_{0:T}|c)}{p_{\mathrm{ref}}(\mathbf{x}_{0:T}|c)}$, and $e^{r(c,\{\mathbf{x}_{0:T}\}_j)}$, we have:

$$\sum_{j=1}^{K} \frac{\hat{p}_\theta(\{\mathbf{x}_{0:T}\}_j|c)}{p_{\mathrm{ref}}(\{\mathbf{x}_{0:T}\}_j|c)} = K\mathbb{E}_{p_{\mathrm{ref}}(\mathbf{x}_{0:T}|c)} \left[ \frac{\hat{p}_\theta(\mathbf{x}_{0:T}|c)}{p_{\mathrm{ref}}(\mathbf{x}_{0:T}|c)} \right] = K,$$

$$\sum_{j=1}^{K} e^{r(c,\{\mathbf{x}_{0:T}\}_j)} = K\mathbb{E}_{p_{\mathrm{ref}}(\mathbf{x}_{0:T}|c)} \left[ e^{r(c,\mathbf{x}_{0:T})} \right] = KZ(c)$$

So that we can simplify the Equation (17):

$$\nabla_\theta \mathcal{L}_{\text{dpo}}(\theta) = \nabla_\theta \mathbb{E}_{c \sim \mathcal{D}_c} \left[ -\frac{1}{K} \sum_{i=1}^{K} \mathbb{E}_{p_{\text{ref}}(\{\mathbf{x}_{0:T}\}_i | c)} \left[ \frac{e^{r(c, \{\mathbf{x}_{0:T}\}_i)}}{Z(c)} \log \frac{\hat{p}_\theta(\{\mathbf{x}_{0:T}\}_i | c)}{\hat{p}^*(\{\mathbf{x}_{0:T}\}_i | c)} \right] \right]$$

$$= \nabla_\theta \mathbb{E}_{c \sim \mathcal{D}_c} \left[ -\mathbb{E}_{p_{\text{ref}}(\mathbf{x}_{0:T} | c)} \left[ \frac{e^{r(c, \mathbf{x}_{0:T})}}{Z(c)} \log \frac{\hat{p}_\theta(\mathbf{x}_{0:T} | c)}{\hat{p}^*(\mathbf{x}_{0:T} | c)} \right] \right]$$

$$= \nabla_\theta \mathbb{E}_{c \sim \mathcal{D}_c} \left[ -\sum_{\mathbf{x}_{0:T} \in \mathcal{D}} p_{ref}(\mathbf{x}_{0:T} | c) \frac{e^{r(c, \mathbf{x}_{0:T})}}{Z(c)} \log \frac{\hat{p}_\theta(\mathbf{x}_{0:T} | c)}{\hat{p}^*(\mathbf{x}_{0:T} | c)} \right]$$

$$= \nabla_\theta \mathbb{E}_{c \sim \mathcal{D}_c} \left[ -\sum_{\mathbf{x}_{0:T} \in \mathcal{D}} \hat{p}^*(\mathbf{x}_{0:T} | c) \log \frac{\hat{p}_\theta(\mathbf{x}_{0:T} | c)}{\hat{p}^*(\mathbf{x}_{0:T} | c)} \right]$$

$$= \nabla_\theta \mathbb{E}_{c \sim \mathcal{D}_c} \left[ \mathbb{D}_{\text{KL}} \left( \hat{p}^*(\mathbf{x}_{0:T} | c) \, \| \, \hat{p}_\theta(\mathbf{x}_{0:T} | c) \right) \right],$$

which completes the proof of Theorem 1.

### A.2 CONVEXITY OF EQUATION (12)

We analyze the convexity by firstly computing the second derivative of Equation (12):

$$f''(u) = \frac{e^u}{(1 + e^{-u})^3} \left[ -ue^u + (\log \frac{1 - \alpha}{\alpha} + 1)e^u + u + 1 - \log \frac{1 - \alpha}{\alpha} \right]. \tag{18}$$

Let $g(u) = -ue^u + (\log \frac{1-\alpha}{\alpha} + 1)e^u + u + 1 - \log \frac{1-\alpha}{\alpha}$, then the sign of $f(u)$ is determined by the sign of $g(u)$, i.e., $\text{sgn}(f(u)) = \text{sgn}(g(u))$). We analyze the derivatives of $g(u)$:

$$g'(u) = e^u \left( \log \frac{1-\alpha}{\alpha} - u \right) + 1, \quad g''(u) = e^u \left( \log \frac{1-\alpha}{\alpha} - u - 1 \right). \tag{19}$$

Clearly, $g''(u) = 0$ when $u_0 = \log \frac{1-\alpha}{\alpha} - 1$ :

- For $u < u_0$, we have $g''(u) > 0$;
- For $u > u_0$, we have $g''(u) < 0$.

Therefore, $g'(u)$ attains its maximum at $u = u_0$. Moreover, we observe that:

$$\lim_{u \to -\infty} g'(u) = 0^+, \quad \lim_{u \to +\infty} g'(u) = -\infty.$$

Thus, by the intermediate value theorem, there exists a unique $u_1 > u_0$ such that $g'(u_1) = 0$. It follows that:

- $g'(u) > 0$ for $u < u_1$,
- $g'(u) < 0$ for $u > u_1$.

Therefore, $g(u)$ attains its global maximum at $u = u_1$. Since $g(u_0) = 2e^{u_0} > 0$, we have:

$$g(u_1) > g(u_0) > 0.$$

Furthermore, observe that $g(0) \equiv 2$, which is strictly positive. Additionally, we have:

$$\lim_{u \to \pm\infty} g(u) = -\infty.$$

Hence, by continuity and the monotonicity of $g(u)$, we conclude that for any $\alpha \in (0, 1)$, there exists an interval $(h_1(\alpha), h_2(\alpha))$ such that

$$h_1(\alpha) < 0 < h_2(\alpha) \quad \text{and} \quad g(u) > 0 \quad \text{for all} \quad u \in (h_1(\alpha), h_2(\alpha)).$$

This proves that $f''(u) > 0$ within this interval.

Moreover, consider the scenario where $\alpha$ is small (e.g., $\alpha \in (0, 0.1]$), in this case, we can view the objective as a cross-entropy loss favoring the class with probability $1 - \alpha$. The corresponding optimal logit is

$$u = \log\left(\frac{1-\alpha}{\alpha}\right).$$

Substituting this into $g(u)$, we obtain

$$g\left(\log\left(\frac{1-\alpha}{\alpha}\right)\right) = \frac{1}{\alpha} > 0.$$

This implies that, under this low-$\alpha$ regime, the interval in which $g(u) > 0$ is not only guaranteed to exist but also sufficiently large to cover the optimizer's likely solution. Therefore, the curvature condition $f''(u) > 0$ holds throughout the practically relevant interval.

### A.3 PROOF OF THEOREM 2

We first start by rearranging $\mathcal{L}_{\text{RLHF}}(p_\theta)$ from Equation (16) :

$$\mathcal{L}_{\text{RLHF}}(\theta) = \mathbb{E}_{c \sim \mathcal{D}_c} \left( \mathbb{E}_{p_\theta(\mathbf{x}_{0:T}|c)}\left[r(c, \mathbf{x}_{0:T})\right] - \beta \mathbb{D}_{\text{KL}}\left[p_\theta(\mathbf{x}_{0:T}|c)\|p_{\text{ref}}(\mathbf{x}_{0:T}|c)\right] \right)$$

$$= \mathbb{E}_{c \sim \mathcal{D}_c} \left( \mathbb{E}_{p_\theta(\mathbf{x}_{0:T}|c)}\left[r(c, \mathbf{x}_{0:T})\right] - \beta \mathbb{E}_{p_\theta(\mathbf{x}_{0:T}|c)}\left[\log \frac{p_\theta(\mathbf{x}_{0:T}|c)}{p_{\text{ref}}(\mathbf{x}_{0:T}|c)}\right] \right)$$

$$= \mathbb{E}_{c \sim \mathcal{D}_c} \left( \beta \mathbb{E}_{\mathbf{x}_{0:T} \sim p_\theta(\cdot|c)}\left[\log e^{r(c, \mathbf{x}_{0:T})}\right] - \beta \mathbb{E}_{\mathbf{x}_{0:T} \sim p_\theta(\cdot|c)}\left[\log \frac{p_\theta(\mathbf{x}_{0:T}|c)}{p_{\text{ref}}(\mathbf{x}_{0:T}|c)}\right] \right)$$

$$= \mathbb{E}_{c \sim \mathcal{D}_c}\mathbb{E}_{\mathbf{x}_{0:T} \sim p_\theta(\cdot|c)}\left[\beta \log\left(\frac{p_{\text{ref}}(\mathbf{x}_{0:T}|c)e^{r(c, \mathbf{x}_{0:T})}}{p_\theta(\mathbf{x}_{0:T}|c)}\right)\right]$$

Utilizing the definition of $\hat{p}^*$, we substitute $p_{\text{ref}}(\mathbf{x}_{0:T}|c)e^{r(c, \mathbf{x}_{0:T})}$ into the expression of $\mathcal{L}_{\text{RLHF}}(\theta)$:

$$\mathcal{L}_{\text{RLHF}}(\theta) = \mathbb{E}_{c \sim \mathcal{D}_c}\mathbb{E}_{\mathbf{x}_{0:T} \sim p_\theta(\cdot|c)}\left[\beta \log\left(\frac{H \cdot \hat{p}^*(\mathbf{x}_{0:T}|c)}{p_\theta(\mathbf{x}_{0:T}|c)}\right)\right]$$

$$= \beta \mathbb{E}_{c \sim \mathcal{D}_c}\left[-\mathbb{D}_{\text{KL}}\left(p_\theta(\mathbf{x}_{0:T}|c)\|\hat{p}^*(\mathbf{x}_{0:T}|c)\right) + H\right].$$

where $H$ is a constant independent of $\theta$. Note that when $\beta = 1$. $p_\theta$ can be also $\hat{p}_\theta$ as the definition. Hence when $\beta = 1$, the final RLHF loss will be :

$$\mathcal{L}_{\text{RLHF}}(\theta) = \mathbb{E}_{c \sim \mathcal{D}_c}\left[-\mathbb{D}_{\text{KL}}\left(\hat{p_\theta}(\mathbf{x}_{0:T}|c)\|\hat{p}^*(\mathbf{x}_{0:T}|c)\right) + H\right], \tag{20}$$

Then we compute the gradient of above loss, as $H$ is a constant independent of $\theta$, finall the RLHF loss will be

$$\nabla_\theta \mathcal{L}_{\text{RLHF}}(\theta) = -\nabla_\theta \mathcal{L}_{\text{DMPO}}(\theta) \tag{21}$$

so that we complete the proof of Theorem 2.

## B EXPERIMENTAL DETAILS

### B.1 DETAILS OF EVALUATION

**Reward Score**. For each generated or edited image $I_1$ and its corresponding target prompt $c$, we compute an automated *reward score*:

$$R(I_1, c) = r(I_1, c),$$

where $r$ is a pretrained vision–language reward model that outputs a scalar alignment score. We directly report the mean reward score over the entire test set for Table 1, Table 7 and Table 2:

$$\text{Reward Score} = \frac{1}{N}\sum_{i=1}^{N} R(I_1^{(i)}, p^{(i)}).$$

**Win rate using Reward Score**. Given a set of $N$ image–prompt pairs $\{(I_1^{(i)}, I_2^{(i)}, p^{(i)})\}_{i=1}^N$, where $I_1^{(i)}$ and $I_2^{(i)}$ are the outputs from method 1 and 2 respectively for prompt $p^{(i)}$, we compute the reward score for each image using a pretrained reward model $r$:

$$R_1^{(i)} = r\big(I_1^{(i)}, p^{(i)}\big), \quad R_2^{(i)} = r\big(I_2^{(i)}, p^{(i)}\big).$$

We then assign a win to the method with the higher reward score on each instance:

$$\delta_i = \begin{cases} 1, & R_1^{(i)} > R_2^{(i)}, \\ 0, & \text{otherwise.} \end{cases}$$

The overall reward-based win rate of method 1 over method 2 is calculated as:

$$\text{WinRate}_{1>2} = \frac{1}{N} \sum_{i=1}^N \delta_i \times 100\%.$$

**Details of user study**.

We randomly sampled a total of 100 prompts—equally drawn from the test set of Pick-a-Pic V2, HPS V2, and Parti-Prompt—and generated one image per prompt by different models for user study. These image–prompt pairs were then presented to human annotators via a custom web-based interface for blind evaluation.

Specifically, we include all baselines in our comparison. We pair the outputs generated by our DMPO-finetuned SD1.5 model with those from the following methods: Diffusion-DPO, Diffusion-KTO, DSPO, and SFT-finetuned SD1.5. This results in a total of 500 image pairs constructed from the 100 prompts.

Each comparison is evaluated by **2 to 3 human annotators**, yielding a total of **1500 judgments**. For each comparison, the annotator is shown a prompt and two images generated by different models, and is asked to answer the three evaluation questions outlined in Section 5.2. We report the final win rates by aggregating the human preferences across all comparisons, as shown in Figure 3.

### B.2 DETAILS OF DATASETS

**Pick-a-Pic** (Kirstain et al., 2023): This comprehensive dataset captures real user preferences from the Pick-a-Pic web platform, where users generate images from text prompts. The collection encompasses more than 500,000 preference examples derived from over 35,000 unique prompts. Each entry consists of a text prompt, two AI-generated images, and user feedback indicating their preferred image or marking cases where no clear preference exists. The dataset incorporates outputs from various generative models, including Stable Diffusion 2.1, Dreamlike Photoreal 2.05, and multiple Stable Diffusion XL configurations, utilizing different classifier-free guidance parameters during generation.

**Parti-Prompts** (Yu et al., 2022): This benchmark dataset features more than 1,600 carefully crafted English prompts designed to evaluate text-to-image model capabilities. The prompts cover diverse categories and present varied challenges, enabling comprehensive assessment of model performance across multiple evaluation criteria.

**HPS V2** (Wu et al., 2023): This preference-based collection contains 98,807 generated images sourced from 25,205 distinct prompts. The dataset structure allows for multiple image generations per prompt, with users selecting their preferred output while other variations serve as negative examples. The distribution includes 23,722 prompts with four associated images, 953 prompts with three images, and 530 prompts paired with two images.

**TEd-bench** (Kawar et al., 2023): Textual Editing Benchmark is a text-to-image editing benchmark that provides pairs of real images and corresponding editing prompts. For example, given a source image of a dog running, the target prompt might be "A cat running on the grass", in which case the model is expected to modify the original image so that it depicts the desired semantics. We evaluate both existing baselines and our proposed method on TEd-bench by applying an automated reward model to score each edited image, and report the average reward as our editing performance metric.

**InstructPix2Pix (Brooks et al., 2023):** This specialized collection focuses on instruction-based image editing capabilities. The dataset enables models to modify existing images according to natural language instructions, such as "make the clouds rainy." The system processes both the editing instruction and the original image to produce the desired modifications. Our experimental evaluation utilized 1,000 test samples from this dataset to assess instruction-following image editing performance. Note that although reward models such as CLIP (Radford et al., 2021) may not comprehensively capture the semantics of instruction-style prompts when scoring edited images, we maintain that if the key terms from the instruction appear faithfully in the generated image, this constitutes a valid comparison.

### B.3 PSEUDO CODE OF DMPO

```python
def loss(model, ref_model, x_w, x_l, c, alpha, beta):
    """
        Calculate the DMPO loss for preferred image pair.
    model: Diffusion model with prompt and timestep conditioning.
    ref_model: Frozen reference model.
    x_w: Preferred latent.
    x_l: Less preferred latent.
    c: Conditioning input (e.g., caption text).
    alpha: smoothing coefficient.
    beta: Regularization strength.
    """
    t = torch.randint(0, 1000)
    noise = torch.randn_like(x_w)
    noisy_x_w = add_noise(x_w, noise, t)
    noisy_x_l = add_noise(x_l, noise, t)
    model_w_pred = model(noisy_x_w, c, t)
    model_l_pred = model(noisy_x_l, c, t)
    ref_w_pred = ref_model(noisy_x_w, c, t)
    ref_l_pred = ref_model(noisy_x_l, c, t)
    model_w_err = (model_w_pred - noise).norm().pow(2)
    model_l_err = (model_l_pred - noise).norm().pow(2)
    ref_w_err = (ref_w_pred - noise).norm().pow(2)
    ref_l_err = (ref_l_pred - noise).norm().pow(2)

    w_diff = model_w_err - ref_w_err
    l_diff = model_l_err - ref_l_err

    inside_term = -1 * beta * (w_diff - l_diff)
    loss_1 = torch.sigmoid(-inside_term) * (torch.logsigmoid(-inside_term
        ) - torch.log(alpha))
    loss_2 = torch.sigmoid(inside_term) * (torch.logsigmoid(inside_term)
        - torch.log(1-alpha))
    loss = loss_1 + loss_2
    return loss
```

## C MORE EXPERIMENTAL RESULTS

### C.1 MORE QUALITATIVE RESULTS

To further demonstrate the effectiveness of DMPO, we provide additional qualitative results on text-to-image alignment and image editing tasks. These results in Figure 5, Figure 6, Figure 7, and Figure 8 clearly show that DMPO consistently produces superior outputs compared to baseline methods.

### C.2 MORE QUANTITATIVE RESULTS

We also conduct quantitative evaluations and analyses for all experiments based on SD1.5 and SDXL. As shown in Table 7, Table 8, Table 2 and Table 5, DMPO demonstrates strong generalization ability,

consistently outperforming other baselines in both reward model scores and win rates across both base models.

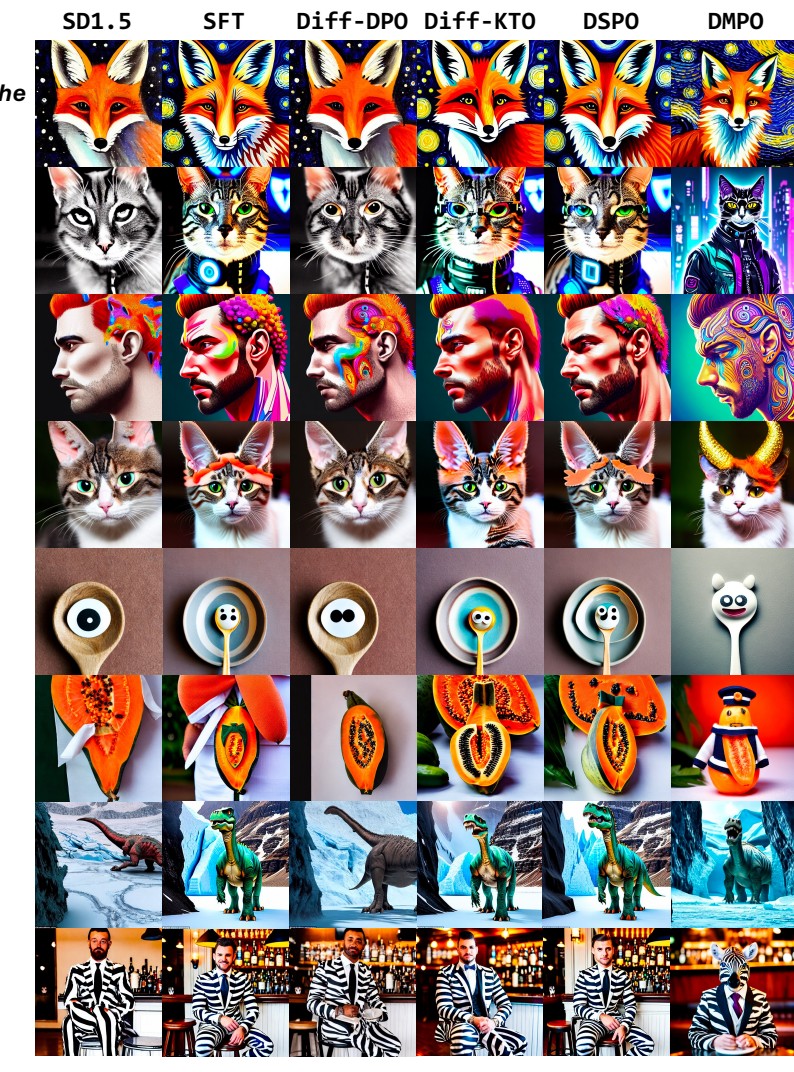

Figure 5: Images generated by different models (based on SD1.5) for various prompts which are selected from Pick-a-Pic V2, Parti-Prompt and HPS V2.

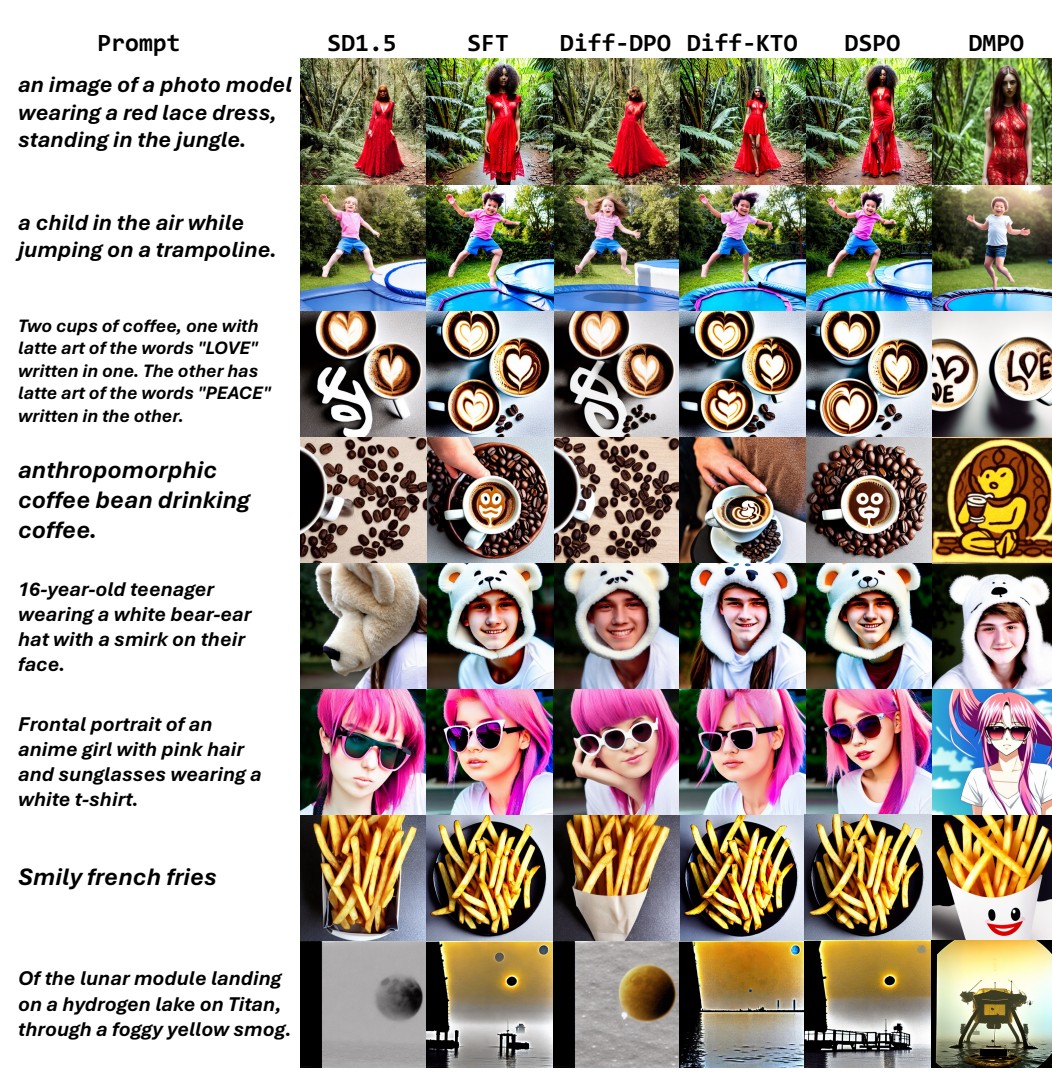

Figure 6: Images generated by different models (based on SD1.5) for various prompts which are selected from Pick-a-Pic V2, Parti-Prompt and HPS V2.

Figure 7: Images generated by different models (based on SDXL) for various prompts which are selected from Pick-a-Pic V2, Parti-Prompt and HPS V2.

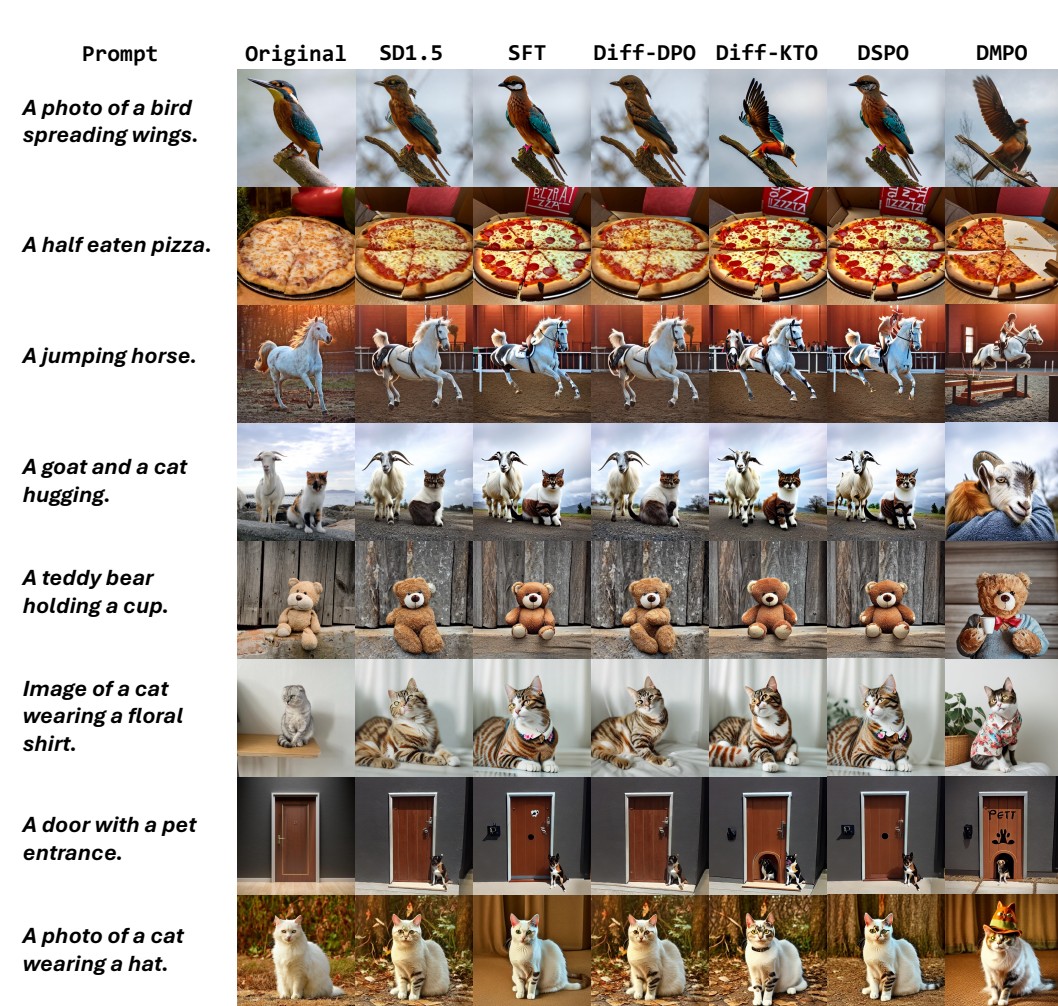

Figure 8: Images edited by different models (based on SD1.5) for various prompts which are selected from TEd-bench.

Table 4: (a) Win rate (%) comparisons on Pick-a-Pic V2, HPS V2 and Parti-Prompt datasets for all baselines versus SD1.5. (b) Win rate comparison between DMPO versus other baselines, win rates that surpass 50 % are in green, below 50 % are in red. For simplicity, "Diff" represents "Diffusion".

| Dataset | Method1 | Method2 | Pick Score | HPS | CLIP | Aesthetics | Image Reward |
|---|---|---|---|---|---|---|---|
| Pick-a-Pic V2 | DMPO | SD1.5 | **82.40** | **84.80** | **65.60** | **75.60** | **80.40** |
| | SFT | SD1.5 | 74.60 | 82.60 | 61.60 | 72.40 | 77.80 |
| | Diff-DPO | SD1.5 | 73.20 | 70.00 | 61.20 | 69.40 | 64.20 |
| | Diff-KTO | SD1.5 | 73.60 | 84.60 | 58.60 | 74.00 | 80.00 |
| | DSPO | SD1.5 | 74.60 | 84.80 | 61.60 | 73.60 | 76.20 |
| | DMPO | SFT | 67.20 | 50.60 | 52.80 | 53.20 | 50.20 |
| | DMPO | Diff-DPO | 73.40 | 77.20 | 56.80 | 64.80 | 68.40 |
| | DMPO | Diff-KTO | 69.00 | 50.80 | 55.20 | 55.20 | 50.20 |
| | DMPO | DSPO | 68.40 | 50.80 | 54.80 | 52.20 | 50.80 |
| HPS V2 | DMPO | SD1.5 | **85.50** | 86.50 | **61.00** | **78.75** | 79.75 |
| | SFT | SD1.5 | 77.00 | 89.00 | 56.00 | 73.75 | **81.50** |
| | Diff-DPO | SD1.5 | 77.50 | 69.25 | 57.75 | 69.5 | 64.50 |
| | Diff-KTO | SD1.5 | 75.25 | **89.50** | 55.00 | 75.75 | 79.25 |
| | DSPO | SD1.5 | 74.50 | 89.00 | 58.00 | 78.25 | 78.75 |
| | DMPO | SFT | 67.25 | 47.50 | 56.00 | 57.75 | 53.50 |
| | DMPO | Diff-DPO | 74.00 | 74.25 | 58.00 | 70.00 | 72.00 |
| | DMPO | Diff-KTO | 68.00 | 47.50 | 56.50 | 55.50 | 53.50 |
| | DMPO | DSPO | 65.75 | 50.00 | 59.00 | 57.00 | 54.00 |
| Parti Prompt | DMPO | SD1.5 | **76.72** | 82.67 | **62.01** | **71.69** | **75.25** |
| | SFT | SD1.5 | 66.36 | 83.88 | 54.35 | 71.51 | 72.37 |
| | Diff-DPO | SD1.5 | 67.16 | 64.86 | 54.84 | 60.66 | 63.11 |
| | Diff-KTO | SD1.5 | 65.56 | 84.38 | 54.41 | 69.97 | 73.86 |
| | DSPO | SD1.5 | 67.30 | **85.17** | 53.31 | 70.22 | 73.71 |
| | DMPO | SFT | 64.64 | 51.31 | 57.23 | 51.69 | 56.00 |
| | DMPO | Diff-DPO | 69.30 | 72.54 | 58.15 | 65.62 | 69.55 |
| | DMPO | Diff-KTO | 67.03 | 48.40 | 59.13 | 55.15 | 55.46 |
| | DMPO | DSPO | 66.12 | 47.14 | 59.25 | 53.49 | 55.88 |

Table 5: (a) Win rate (%) comparisons on Pick-a-Pic V2, HPS V2 and Parti-Prompt datasets for all baselines (fine-tuned on SDXL) versus SDXL. (b) Win rate (%) comparison between DMPO versus other baselines, win rates that surpass 50 % are in green, below 50 % are in red. For simplicity, "Diff" represents "Diffusion".

| Dataset | Method1 | Method2 | Pick Score | HPS | CLIP | Aesthetics | Image Reward |
|---------|---------|---------|-----------|-----|------|-----------|-------------|
| Pick-a-Pic V2 | DMPO | SDXL | **75.20** | **80.20** | 61.60 | 48.00 | **69.80** |
| | SFT | SDXL | 54.20 | 67.50 | 53.50 | 43.20 | 61.80 |
| | Diff-DPO | SDXL | 74.50 | 79.00 | **61.80** | 51.00 | 69.00 |
| | MAPO | SDXL | 55.60 | 68.00 | 51.20 | 66.60 | 64.20 |
| | DSPO | SDXL | 56.30 | 69.20 | 52.60 | 44.80 | 60.50 |
| | DMPO | SFT | 65.80 | 72.00 | 60.20 | 53.20 | 66.20 |
| | DMPO | Diff-DPO | 51.50 | 67.00 | 50.80 | 45.40 | 61.20 |
| | DMPO | MAPO | 66.20 | 74.00 | 61.60 | 35.80 | 64.60 |
| | DMPO | DSPO | 63.40 | 73.40 | 63.40 | 50.30 | 63.20 |
| HPS V2 | DMPO | SDXL | 70.75 | **87.25** | **63.75** | 47.75 | **71.25** |
| | SFT | SDXL | 54.75 | 70.00 | 53.00 | 61.25 | 60.00 |
| | Diff-DPO | SDXL | **70.50** | 81.25 | 61.75 | 59.25 | 68.75 |
| | MAPO | SDXL | 54.00 | 79.50 | 52.00 | **70.00** | 62.75 |
| | DSPO | SDXL | 53.25 | 71.25 | 55.25 | 57.35 | 64.75 |
| | DMPO | SFT | 60.75 | 65.50 | 58.75 | 50.00 | 63.50 |
| | DMPO | Diff-DPO | 51.25 | 72.00 | 53.00 | 43.00 | 52.50 |
| | DMPO | MAPO | 62.25 | 75.25 | 59.50 | 38.50 | 66.75 |
| | DMPO | DSPO | 63.75 | 73.00 | 59.75 | 48.25 | 62.25 |
| Parti Prompt | DMPO | SDXL | **68.72** | **79.96** | **61.46** | 51.69 | **71.75** |
| | SFT | SDXL | 50.44 | 66.58 | 50.75 | 52.14 | 56.69 |
| | Diff-DPO | SDXL | 66.89 | 76.89 | 60.64 | 54.96 | 69.30 |
| | MAPO | SDXL | 49.94 | 67.34 | 50.80 | **69.24** | 59.44 |
| | DSPO | SDXL | 52.44 | 65.17 | 53.31 | 53.22 | 58.71 |
| | DMPO | SFT | 62.22 | 66.31 | 59.23 | 50.69 | 53.46 |
| | DMPO | Diff-DPO | 56.07 | 61.89 | 52.77 | 45.62 | 52.14 |
| | DMPO | MAPO | 64.22 | 70.22 | 62.01 | 44.21 | 66.24 |
| | DMPO | DSPO | 60.12 | 63.08 | 58.25 | 51.49 | 57.52 |

Table 6: (a) Win rate (%) comparisons on Pick-a-Pic V2, HPS V2 and Parti-Prompt datasets for all baselines (fine-tuned on SD3.5-M) versus SD3.5-M. (b) Win rate (%) comparison between DMPO versus other baselines, win rates that surpass 50 % are in green, below 50 % are in red. For simplicity, "Diff" represents "Diffusion".

| Dataset | Method1 | Method2 | Pick Score | HPS | CLIP | Aesthetics | Image Reward |
|---------|---------|---------|-----------|-----|------|-----------|-------------|
| Pick-a-Pic V2 | DMPO | SD3.5-M | **65.20** | **70.10** | **58.40** | **54.40** | **59.20** |
| | Diff-DPO | SD3.5-M | 61.50 | 66.00 | 53.80 | 52.00 | 56.00 |
| | DMPO | Diff-DPO | 54.50 | 59.00 | 59.80 | 53.40 | 57.20 |
| HPS V2 | DMPO | SD3.5-M | **63.75** | **77.50** | **63.75** | **52.75** | **68.25** |
| | Diff-DPO | SD3.5-M | 58.50 | 71.25 | 57.75 | 49.25 | 62.75 |
| | DMPO | Diff-DPO | 52.75 | 64.00 | 55.00 | 51.00 | 56.50 |
| Parti Prompt | DMPO | SD3.5-M | **63.45** | **71.29** | **63.23** | **52.87** | **67.54** |
| | Diff-DPO | SD3.5-M | 59.11 | 66.49 | 62.64 | 50.96 | 64.30 |
| | DMPO | Diff-DPO | 55.37 | 58.92 | 55.74 | 49.24 | 56.48 |

Table 7: Reward Score comparisons on TEd-bench and Instructpix2pix datasets for all baselines versus SD1.5, best results are in **boldface**. For simplicity, "Diff" represents "Diffusion".

| Dataset | Method | Pick Score(↑) | HPS(↑) | CLIP(↑) | Aesthetics (↑) | Image Reward (↑) |
|---------|--------|---------------|--------|---------|----------------|------------------|
| TEd-bench | SD1.5 | 0.2165 | 0.2743 | 0.3043 | 5.4194 | -0.0078 |
| | DMPO | **0.2218** | **0.2796** | **0.3301** | 5.5982 | **0.5619** |
| | SFT | 0.2185 | 0.2789 | 0.3061 | **5.6185** | 0.1869 |
| | DPO | 0.2182 | 0.2756 | 0.3065 | 5.4676 | 0.0601 |
| | KTO | 0.2190 | 0.2795 | 0.3125 | 5.6142 | 0.3298 |
| | DSPO | 0.2183 | 0.2787 | 0.3094 | 5.6022 | 0.1822 |
| Instructpix2pix | SD1.5 | 0.2044 | 0.2561 | 0.2557 | 5.4923 | -0.4589 |
| | DMPO | **0.2090** | **0.2618** | **0.2879** | 5.4849 | **-0.0151** |
| | KTO | 0.2073 | 0.2612 | 0.2700 | 5.7312 | -0.0623 |
| | DPO | 0.2054 | 0.2572 | 0.2613 | 5.5066 | -0.3727 |
| | SFT | 0.2076 | 0.2610 | 0.2680 | **5.8001** | -0.0986 |
| | DSPO | 0.2076 | 0.2611 | 0.2687 | 5.7972 | -0.0825 |

Table 8: (a) Win rate (%) comparisons about imaging editing on TEd-bench datasets for all baselines (fine-tuned on SD1.5) versus SD1.5. (b) Win rate (%) comparison between DMPO versus other baselines, win rates that surpass 50 % are in green, below 50 % are in red. For simplicity, "Diff" represents "Diffusion".

| Dataset | Method1 | Method2 | Pick Score | HPS | CLIP | Aesthetics | Image Reward |
|---------|---------|---------|------------|-----|------|------------|--------------|
| TEd-bench | DMPO | SD1.5 | **80.00** | 88.00 | **71.00** | **84.00** | **85.00** |
| | SFT | SD1.5 | 67.00 | 83.00 | 46.00 | 83.00 | 71.00 |
| | DPO | SD1.5 | 75.00 | 69.00 | 56.00 | 62.00 | 55.00 |
| | KTO | SD1.5 | 64.00 | **94.00** | 54.00 | 83.00 | 69.00 |
| | DSPO | SD1.5 | 59.00 | 81.00 | 50.00 | 84.00 | 71.00 |
| | DMPO | SFT | 64.00 | 51.00 | 71.00 | 48.00 | 66.00 |
| | DMPO | DPO | 74.00 | 77.00 | 70.00 | 63.00 | 78.00 |
| | DMPO | KTO | 60.00 | 48.00 | 64.00 | 50.00 | 53.00 |
| | DMPO | DSPO | 67.00 | 50.00 | 68.00 | 51.00 | 69.00 |
| Instructpix2pix | DMPO | SD1.5 | **81.30** | 80.60 | **75.50** | 58.50 | **78.90** |
| | SFT | SD1.5 | 77.00 | 85.80 | 64.90 | 79.30 | 78.00 |
| | DPO | SD1.5 | 68.50 | 67.70 | 60.50 | 54.50 | 61.10 |
| | KTO | SD1.5 | 73.40 | **84.70** | 66.10 | **72.80** | 74.80 |
| | DSPO | SD1.5 | 77.40 | 85.10 | 64.80 | 79.50 | 78.10 |
| | DMPO | SFT | 63.30 | 51.60 | 69.70 | 40.30 | 52.70 |
| | DMPO | DPO | 75.90 | 76.60 | 73.90 | 54.40 | 72.40 |
| | DMPO | KTO | 63.60 | 52.20 | 66.50 | 43.90 | 51.10 |
| | DMPO | DSPO | 61.40 | 51.10 | 68.20 | 40.60 | 49.00 |

## C.3 TRAINING STABILITY

To assess the stability of our optimization procedure, we examine the training loss under two different $\alpha$ values, as shown in Figure 9. In both cases, the loss decreases smoothly and converges within a narrow range, demonstrating stable and reliable optimization dynamics throughout training.

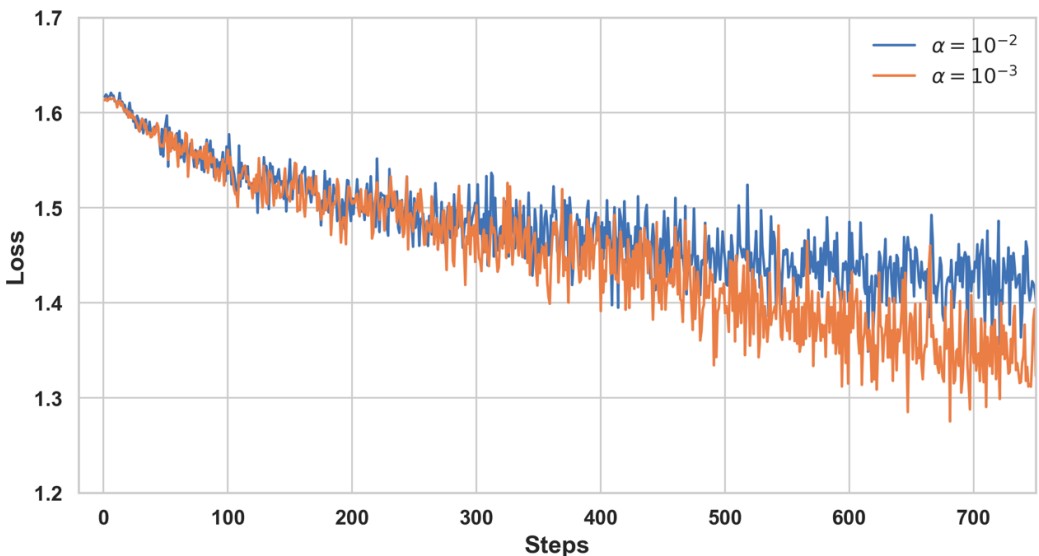

Figure 9: Training loss curves for $\alpha = 10^{-2}$ and $\alpha = 10^{-3}$. In both configurations, the loss exhibits smooth decay and stable convergence, confirming the robustness of the training process.

## C.4 ABLATION STUDY

We conduct ablation studies with $\alpha$ and $\beta$ to understand the sensitivity and behavior of our alignment objective under a consistent setting: we used SD1.5 as the base model, fine-tuned with PickScore on the Pick-a-Pic V2 training dataset, and evaluated on both Pick-a-Pic V2 and HPS V2 using PickScore as the metric.

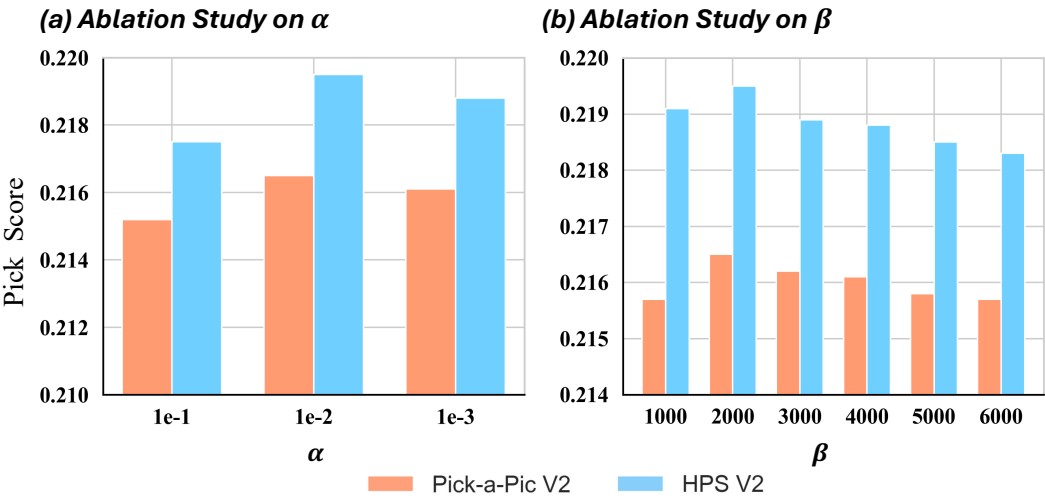

Figure 10: **Ablation study** on parameters $\alpha$ and $\beta$ in DMPO based on SD1.5, evaluated on the Pick-a-Pic V2 and HPS V2 test sets. (a) Effect of $\beta$: **With $\alpha$ fixed at 0.01**, model performance first increases and then decreases as $\beta$ grows. (b) Effect of $\alpha$: **With $\beta$ fixed at 2000**, performance also first increases and then decreases as $\alpha$ increases.

Figure 10a illustrates the performance as $\beta$ increases. As $\beta$ decreases, the optimization objective degenerates into a pure reward function, leading to a drop in performance. Conversely, as $\beta$ increases, the KL-divergence penalty becomes overly restrictive, greatly limiting the model's capacity to adapt. Figure 10b illustrates the performance as $\alpha$ increases. Regarding the smoothing coefficient $\alpha$, we set $\alpha$ to be a small positive number since it represents the probability of the less preferred sample to avoid numerical instability. Therefore, we set $a \in \{0.1, 0.01, 0.001\}$ for ablation study. As shown in Appendix A, we observe that as $\alpha$ decreases, the second derivative of the objective $f(u)$ in Equation (12) increases for $u > 0$, resulting in a looser upper bound. This may weaken alignment precision. In particular, when $\alpha$ is very small, the reward term becomes smoothed by the coefficient $\alpha$, causing the model to occasionally favor less-preferred samples in preference pairs.

