# OpenReview forum: "Divergence Minimization Preference Optimization for Diffusion Model Alignment"
_ICLR.cc/2026/Conference — ICLR 2026 Conference Withdrawn Submission_

### Official Review · Reviewer_xwem · 2025-10-27

**Soundness:** 3
**Presentation:** 3
**Contribution:** 2
**Rating:** 4
**Confidence:** 5

**Summary:**

The paper addresses the well-known issue of mean-seeking behavior inherent to the forward KL objective when aligning diffusion models. The authors propose an alternative formulation based on the reverse KL objective, thereby inducing mode-seeking behavior that aims to reduce undesirable generations. The derivation is rigorous and well-motivated. Experimental results, including both standard synthetic preference benchmarks and a user study, demonstrate the proposed method’s advantages in terms of overall preference, visual appeal, and prompt alignment.

**Strengths:**

1. The motivation is clear and well-founded, as the forward-KL objective is known to induce mean-seeking behavior.

2. The proposed approach yields promising results and has the potential to serve as a plug-and-play replacement for the standard DPO objective in production-ready pipelines.

3. The paper provides rigorous derivations, carefully establishing and demonstrating the equivalence between the proposed formulation and the original RLHF objective.

**Weaknesses:**

1. While the DPO objective is known to “cut” gradients when the model deviates from the reference, the proposed objective appears to exhibit a similar behavior. However, the paper does not include an analysis of this property. A gradient-flow or stability analysis would be valuable for understanding why the proposed loss performs well, whether it is more or less stable than DPO, and how sensitive it is to hyperparameter choices.

2. The authors argue that mode-seeking behavior is preferable to mean-seeking, yet it remains unclear how this affects model diversity during training. Objectives such as DPO or other RLHF variants are known to risk mode collapse in practice, leading to repetitive or overly similar generations. It would strengthen the paper to analyze whether the proposed loss mitigates or exacerbates this issue, particularly when moving beyond in-domain prompts.

3. Building on the previous point, the evaluation of out-of-distribution generalization is somewhat limited. While the paper includes results on the PartiPrompts dataset, the use of only 100 prompts provides an insufficient assessment of generalization. Given that alignment methods ideally should generalize to unseen domains and cannot rely on exhaustive reward coverage, a more extensive OOD evaluation would significantly improve the empirical analysis.

**Questions:**

1. Could you provide an analysis of how strongly your loss function “cuts” gradients compared to DPO?

2. How does the proposed objective affect model diversity during training? Have you observed any signs of mode collapse or reduced sample variety?

3. Could you expand the evaluation to include a more comprehensive out-of-distribution analysis?

4. Do you expect particular hyperparameters to control the trade-off between stability, diversity, and alignment strength? If so, could you provide guidance on how to tune them effectively?

5. What hyperparameters were used for training the Diffusion-DPO checkpoints? The reported win rates and results compared to Diff-DPO appear somewhat questionable and may require additional hyperparameter tuning. Moreover, HPS and PickScore metrics are typically highly correlated with ImageReward, yet Diff-DPO shows unusually low results in Table 1.

I would be happy to reconsider and increase my score if the authors address the questions and analyses raised above in their rebuttal. The paper is well-executed, but addressing these concerns is essential for the community and would enhance the paper’s clarity and value as a contribution to the field. I understand the tight timeline for the rebuttal and appreciate any clarifications or additional insights the authors can provide within that scope.

---

### Official Review · Reviewer_P6x3 · 2025-10-31

**Soundness:** 2
**Presentation:** 3
**Contribution:** 2
**Rating:** 2
**Confidence:** 4

**Summary:**

This paper introduces “Divergence Minimization Preference Optimization (DMPO)”, a method for aligning diffusion models with human preferences by minimizing the reverse KL divergence between the learned policy and the theoretical optimal policy. The authors provide a theoretical analysis, arguing that existing methods like Diffusion-DPO effectively minimize a forward KL, leading to suboptimal "mean-seeking" behavior, whereas their reverse KL approach promotes "mode-seeking" for more precise alignment.

**Strengths:**

1.The core insight of applying a “reverse KL divergence objective” to diffusion model alignment is well-motivated. The paper effectively critiques the limitations of existing DPO-style methods from a distribution-matching perspective, providing a fresh and principled viewpoint on the alignment problem.
2.Theorem 2 establishes that DMPO's optimization direction aligns with the original RLHF objective under certain conditions, lending strong theoretical justification to the proposed approach. The convexity analysis of the loss function is also rigorous.

**Weaknesses:**

1. While reverse KL divergence does have its advantages, it does not necessarily mean it is always better than forward KL divergence; both have their own characteristics. Although Sections 3.1 and 3.2 provide extensive textual descriptions, there is a lack of specific theoretical validation in the context of alignment issues, rather than just textual explanations. I have not seen more detailed analysis on this. In Section 3.3, only the correlation with RLHF is mentioned, but the loss function of Diffusion-DPO is also derived from RLHF. Moreover, in the loss function of DMPO, the forward distribution is used to estimate the reverse process, which is the same as in Diffusion-DPO.
2. The paper would be strengthened by a brief discussion or comparison of the computational cost of DMPO versus baselines (e.g., Diffusion-DPO, DSPO).  This is a practical concern for researchers and practitioners looking to adopt the method.
3. There is a lack of comparison with too many baselines. The methods compared by the authors are mostly earlier works, with insufficient comparison to recent works like [1]SPO(SD1.5 SDXL), [2]DDIM-InPO(SD1.5 SDXL), [3]Flow-GRPO(SD3.5-Medium). These baselines (checkpoints) are open-source.
 [1]Aesthetic Post-Training Diffusion Models from Generic Preferences with Step-by-step Preference Optimization. CVPR2025
 [2]InPO: Inversion Preference Optimization with Reparametrized DDIM for Efficient Diffusion Model Alignment. CVPR2025
 [3]Flow-GRPO: Training Flow Matching Models via Online RL

**Questions:**

1. Based on the results shown in Figure 1, for instance, could reverse KL divergence affect the diversity of diffusion-generated outputs?

2. A dedicated paragraph discussing the limitations of DMPO would enhance the paper. For instance,are there any failure modes observed?

3. Is the training time, memory footprint, or inference time comparable?

4. I think the metrics on the reward function are not very accurate and quite limited. Why not test the method's effectiveness on benchmarks like T2I-CompBench and GenVal?

---

### Official Review · Reviewer_UGqu · 2025-10-31

**Soundness:** 3
**Presentation:** 3
**Contribution:** 3
**Rating:** 4
**Confidence:** 4

**Summary:**

This paper studies preference alignment for diffusion models from a divergence-minimization viewpoint. The authors argue that most current preference optimization methods for diffusion models implicitly behave like mean-seeking objectives and can get stuck in suboptimal solutions. To address this, they propose Divergence Minimization Preference Optimization (DMPO), which directly minimizes the reverse KL and, according to their analysis, recovers the same optimization direction as RL-style objectives but in a more principled and stable way. They provide theoretical justification for why reverse-KL-based alignment is better suited for capturing sharp, human-preferred modes, and they validate the method on multiple diffusion backbones and test sets. Empirically, DMPO is reported to consistently match or surpass existing alignment techniques and to achieve the best PickScore across all settings, including human evaluation. Overall, the paper claims that DMPO offers a cleaner, theory-backed route to preference alignment for diffusion models, narrowing the gap between RL-style formulations and practical diffusion fine-tuning.

**Strengths:**

1. The paper is well written and easy to follow.

2. Research on Human preference alignment of T2I generation are important. Mean-seeking issues are really important for model training.

3. The authors conduct extensive experiments to verify the effectiveness of their method.

**Weaknesses:**

1. The problem of mean-seeking is widely studied in LLMs like EXO, f-PO as mentioned in the related works. And there are also some previous works about the derivation of adopting DPO algorithms on LLMs to a chain of Markov transitions. Therefore, I think the contribution of this paper is limited.

2. Do authors study different divergence optimization methods like f-PO on T2I diffusion models?

3. It's better for authors to provide time complexity analysis or training time comparison to further verify the effectiveness of their method.

**Questions:**

Please refer to weaknesses part.

---

### Official Review · Reviewer_wGg7 · 2025-11-03

**Soundness:** 2
**Presentation:** 2
**Contribution:** 2
**Rating:** 2
**Confidence:** 4

**Summary:**

The paper proposes **Divergence Minimization Preference Optimization (DMPO)**, a method for aligning diffusion models by minimizing the reverse KL divergence. The authors argue that existing preference optimization approaches suffer from suboptimal, mean-seeking behavior and that DMPO provides a more principled and theoretically grounded solution. The paper presents both theoretical analysis and empirical experiments showing that DMPO achieves superior or comparable performance to existing techniques on human preference alignment benchmarks.

**Strengths:**

* The paper is well-written overall and proposes to solve the diffusion alignment problem through Reverse-KL perspective.
* The experimental evaluation is reasonably comprehensive, covering multiple datasets and baselines.

**Weaknesses:**

* The authors’ discussion of **mode-seeking vs. mode-covering** behavior is confusing, incorrect, and internally inconsistent. In Section 3.2, they claim that Diffusion-DPO approximately corresponds to Forward KL and then state it enforces mean-seeking behavior, but later describe it as mode-covering. The logical flow and terminology need clarification.
* In abstract, the claim that existing methods are trapped in *mode-seeking* optimization contradicts the introduction of DMPO as using **reverse KL**, which is itself known to be **mode-seeking**. This inconsistency undermines the conceptual framing of the paper.
* While reverse KL can indeed align with reward-chasing behavior in RL, it is also well known to cause **mode collapse and loss of diversity**. The authors have not analyzed or discussed these potential drawbacks.
* The **experimental improvements appear incremental**, and the **visual results** (e.g., Fig. 7) do not convincingly demonstrate qualitative advantages — most methods look similar.
* Some **reported results** (e.g., HPS = 27.5 for DSPO) do not match the values reported in the original DSPO paper (≈ 27.8), suggesting possible inconsistencies in evaluation or implementation details.

**Questions:**

In theorem 1, the policy $\hat{p_\theta}$ is just proportional to $p_\theta$ with $p_{ref}$ inside, so it not clean to claim Diffusion-DPO is a similar Forward-KL optimziation.

---

### Note · Authors · 2025-11-14

**Comment:**

Thank you for the reviews. We will provide a brief response to the reviewers’ questions soon.

**Withdrawal Confirmation:**

I have read and agree with the venue's withdrawal policy on behalf of myself and my co-authors.